# And growth on form? How tissue expansion generates novel shapes, colours and enhance biological functions of Turing colour patterns of Eukaryotes

Pierre Galipot[1,2,3]*

1 Institut de Systématique, Évolution, Biodiversité (ISYEB), Muséum National d'Histoire Naturelle, CNRS, Sorbonne Université, EPHE, Université des Antilles, Paris, France, 2 Department of Biological Sciences, Graduate School of Science, The University of Tokyo, Tokyo, Japan, 3 UMR CNRS 6553, Ecosystèmes-Biodiversité-Evolution, OSUR, Université de Rennes 1, Bâtiment, Rennes Cedex, France

* pierregalipotpro@gmail.com

**Data Availability Statement:** The data underlying the results presented in the study are available in a Biostudies repository: https://www.ebi.ac.uk/

## Abstract

Evidenced in zebrafishes skin and *Mimulus* petal, Turing-like mechanisms are probably responsible for many periodic color patterns of Eukaryotes. They are characterized by the mathematical relationships linking their cellular or molecular actors, the periodicity and the geometrical range of the patterns they produce: spots, stripes or mazes. Nevertheless, some periodic patterns such as leopard iconic rosettes required additional ingredients to explain their formation. Growth being the main candidate, we extensively explore its multiple facets, at the Eukaryotes scale. We show that far beyond the particular feline coat pattern, putative-growth Turing color patterns are present in many diverse lineages of plants and animals and seem absent in Fungi and unicellular lineages. Using models, we show the many ways growth can induce new shapes and colors, and that putative-growth pattern locations correlates with tissue hot spots of growth, suggesting the latter as the underlying mechanism. By reverse reasoning, we show that growth effects could reveal crucial information about pattern formation. We show how putative growth patterns can contribute to influence organisms visibility, thereby improving camouflage or aposematism. Our results demonstrate the range of morphogenetic roles that tissue expansion can take, by interacting with a scale-sensitive mechanism, here Turing-like patterning. Considering this extensive overview of its biological importance, both qualitatively and quantitatively, links between growth and form might more than ever needed to be explored.

## Introduction

More than a century ago, D'Arcy Thompson questioned the mathematical and biological relationships between growth and form [1]. Here, we propose that the former could acquire a morphogenetic role when the latter is sensitive to scale, by exploring their biological importance in Turing color patterns of Eukaryotes.

biostudies/studies/S-BSST1375 DOI: 10.6019/S-BSST1375.

**Funding:** The author(s) received no specific funding for this work.

**Competing interests:** The authors have declared that no competing interests exist.

Repeated color patterns are known to allow countless ecological roles such as be more visible to congeners, for partner recognition [2] or communication [3], or to other species, in an attractively manner for pollination [4] or repellingly for aposematism [5]. Repeated color patterns may also avoid detection or recognition, in camouflage [6].

Turing-like systems are composed of a couple of activator and inhibitor in their simplest molecular form [7]. After years of mathematical formalization [8] and modelling [9], compatible patterning systems were described *in vivo* [10] and might be responsible for many biological repeated structures called Turing-like patterns. Characterized by their periodicity and the range of repeated motifs shapes (spots, stripes, mazes and not much more) in the classic Turing-like systems [11], they form for example the vertebrate digits [12], and fingerprints [13] or the xylem openwork patterns of plants [14]. For color patterns, their cellular and molecular bases have been unveiled in zebrafishes [15] and recently in plants of the genus *Mimulus* [16], while currently being explored in more and more other eukaryotic taxa, like the Felidae [17].

The intriguing and iconic leopard (*Panthera pardus*) pattern, characterized by brown rosettes with an orange heart on a light-yellow background, is not under the scope of a classic Turing pattern [11]. Consequently, the leopard color pattern has been the focus of attempts to explain its formation [18, 19], with recent models impressive in both their realism and their biological plausibility [20]. Most of those models perfectly reproduced leopard rosettes by adding the most probable missing ingredient: tissue growth during pattern formation. Other types of missing ingredients have been described, like complex pre-patterns [21, 22]. But what about beyond leopard famous color pattern? Is there a biodiversity of putative-growth Turing color patterns (hereafter called PGTCPs), and what are their qualitative and quantitative characteristics?

To answer this question, we sought to measure in the broadest possible way the importance that growth could have in the formation and the diversity of the periodic color patterns in the Eukaryotes, beyond the biological curiosity of this feline coat.

For a reason of accuracy, we therefore refer to *putative classic Turing-like color patterns*, sometimes simplified as *'classic'* and *putative growth Turing-like color patterns*, abbreviated as PGTCPs. 'Turing-like' refers to the fact that Turing systems can be of many biological scales and natures [11]. Here, we consider a particular color pattern as a PGTCP when it possesses all of these characteristics: i) periodic, at least at the local scale, ii) non-based on a pre-existing periodicity (e.g. body segments or petals), iii) with at least three motifs (e.g. three stripes or rosettes) and iv) with at least one motif not belonging to the classical range of Turing geometries (dots, stripes, mazes), i.e. line-and-band alternations, rosettes for example.

Five research themes were addressed in this study: i) What are the specific geometric and color diversities of PGTCPs and what is their biodiversity in Eukaryotes? ii) By which ways growth can influence the final shapes and colors of Turing patterns and how can it be modelled? iii) Since no functional study has so far directly proven the involvement of growth in PGTCPs, what clues are pointing to it? iv) Can PGTCPs phenotypes be geometrically interpreted and give important information on the Turing system itself and/or on the processes underwent by the tissues during morphogenesis? Finally, v) could PGTCPs improve or provide new biological functions compared to putative classic Turing-like color patterns?

We then discussed the factors controlling the potential rarity of PGTCPs observed in some taxa, whether it is real (linked to ecological, evolutionary or mechanical reasons) or artificial (linked to observation biases).

In summary, we have shown that the diversity of PGTCPs extends far beyond leopards and a few other animals, to thousands of species, including angiosperms. In addition, while some of the effects of growth were already known (like rosette formation), we used simulations to

show other possible effects that had not yet been described and were biologically plausible. We then produced a large number of concordant clues of different kinds, showing the very high probability of growth as the cause of PGTCPs. As the great diversity of PGCTPs may suggest potential biological functions, we have described and discussed several of them, with supporting arguments.

## Results

### Classic and putative growth Turing-like color patterns (PGTCPs) are common and diversified in Angiosperms and Animalia but seem absent in Fungi and other lineages of Eukaryotes

While the leopard has been for a long time the only species studied for its PGTCP, some other animal species have been proposed as candidates during the last years, such as the whale shark [23], the ground squirrel *Ictidomys tridecemlineatus*, or some species of *Dendrobates* frogs [20]. In order to provide a most comprehensive picture possible of the presence of these motifs, we have extended our research to all Eukaryotes. A first large-scale screening showed that PGTCPs seem absent from certain lineages of multicellular Eukaryotes, in particular in Plantae other than Angiosperms (probably due to the absence of biotic pollination), as well as in Fungi (possibly due to the presence of a pre-existing mechanism of repeated pattern formation by physical cracking [24]) and to unicellular lineages of Eukaryotes (where visual signals might be less relevant at the microscopic scale). We then restricted our study to Metazoans and Angiosperms, trying to find from multiple sources at least one example of a species bearing PGTCPs for each of the 620 orders of animals and 422 families (or suitable taxonomic rank when family did not exist) of Angiosperms. The PGTCPs were classified in an exhaustive list of four categories, which could be combined: line-and-dot alternation, mixed colors, rosettes (including mazes of mazes) and intermediate bands (Fig 1A). The complete results are available in a database accessible at the following URL (www.eukolorpatterns@cnrs.fr).

Fig 1B and 1E show the mapping of the different types of patterns (classic or PGTCPs) onto the phylogenetic trees of animals and Angiosperms. In the former, PGTCPs were found in 62 orders (10% of the total), while in 268 orders (43%) only classic patterns with or without suspicion of PGTCPs were found. The remaining 290 orders (47%) did not seem to show any Turing-like pattern, whether classic or PGTCPs.

For Angiosperms, where to our knowledge no species had been explicitly described with PGTCPs, we found 38 families with at least one species displaying PGTCPs, 22 families with potential PGTCPs, plus 5 families with only classic Turing patterns, 33 families with potential classic Turing patterns and finally 339 families for which no species displaying any putative Turing color pattern was found. The criteria for the classification into each category (PGTCPs, Maybe PGTCPs, classic Turing, Maybe classic Turing, none) are detailed in S1 and S3 Fig.

In both Angiosperms and animals, PGTCPs have been found in widely separated orders and families, suggesting several independent evolutions. Some subgroups such as the Lamiales or Actinopterygii appear to harbor a high proportion of PGTCPs; by contrast, some taxa with high species diversity and high coloration diversity seem to harbor fewer PGTCPs than expected, such as insects or birds. We proposed possible ecological, evolutionary and/or biological constraints explanations for these unexpected scarcities in the Discussion part and in Table 1.

Finally, we carried out a quasi-exhaustive species-scale survey of color patterns in two groups that are iconic in terms of color patterns and where species are well described in the literature: mammals (6649 species, Fig 1C) and orchids (29573 species, Fig 1D). Among the former, 10% of the species were found to clearly display putative Turing-like patterns (541

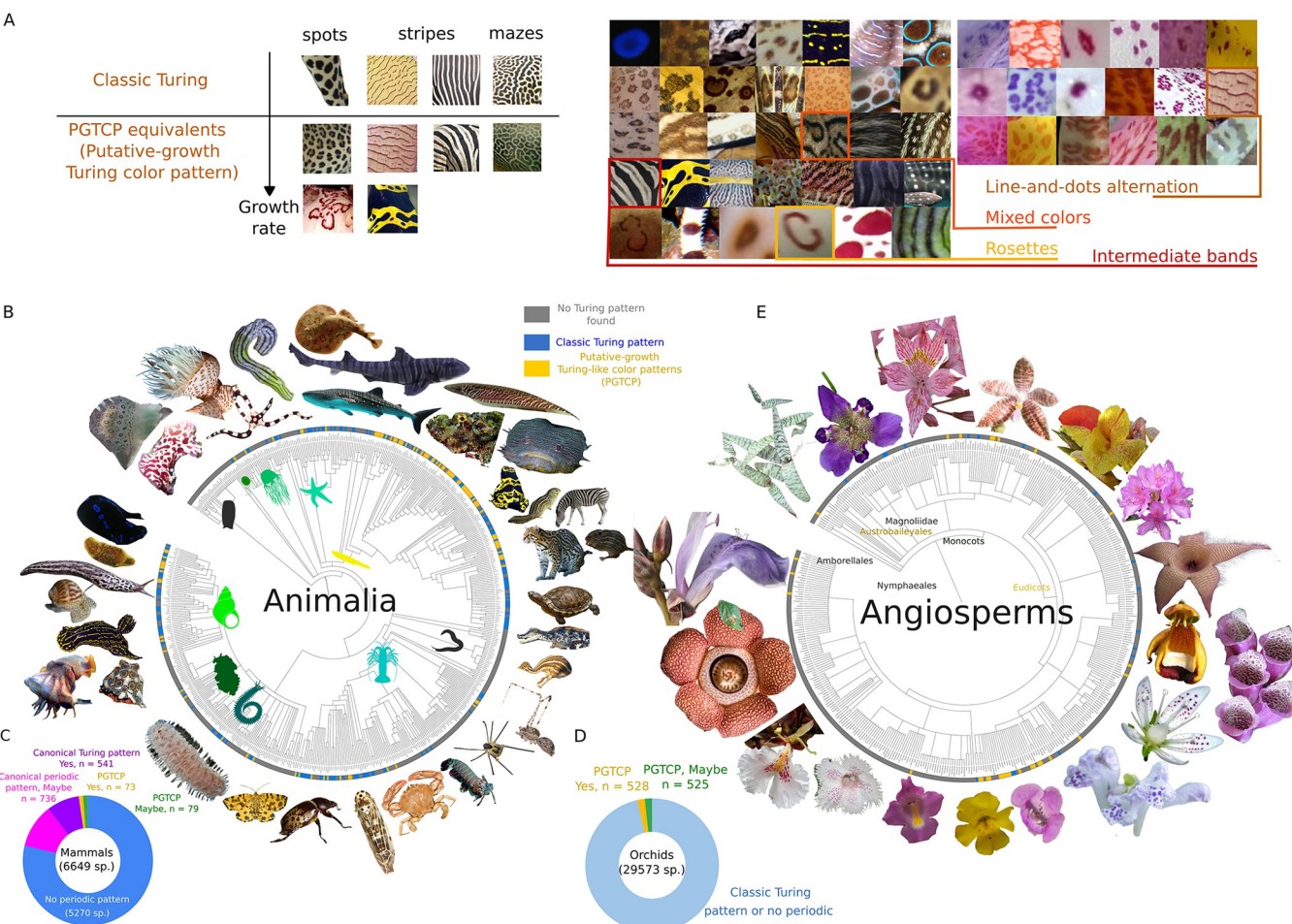

**Fig 1. Diversity of growth-putative Turing colour patterns (PGTCPs) of Eukaryotes.** (A) Left: classification of Turing colour patterns by their fundamental geometry (spots, stripes, mazes) and the presence (PGTCP) or absence (classic) of growths effects. Right: details of PGTCPs in Animalia (left) and Angiosperms (right), divided into an exhaustive list of four categories: line-and-dots alternation, mixed colour, rosettes, intermediate bands. (**B**) Occurrences of classic Turing patterns and PGTCPs in all Animalia orders mapped on Animalia phylogenetic tree. Grey leaf: no Turing pattern found in this order, blue leaf: only classic Turing pattern found in this order, yellow leaf: PGTCP found in at least one species. (**C**) Proportions of classic Turing patterns and PGTCPs for mammals exhaustive focus. (**D**) Proportions of classic Turing patterns and PGTCPs for orchids exhaustive focus. (**E**) Occurrences of classic Turing patterns and PGTCPs in all Angiosperms families. (**B to E**) The criteria for the classification into each category (PGTCPs, Maybe PGTCPs, classic Turing, Maybe classic Turing, none) are detailed in S3 Fig. Picture sources: prilfish,2009,https://commons.wikimedia.org/wiki/File:Chelidonura_livida,_Dahab.jpg, Rickard Zerpe, 2014, https://fr.m.wikipedia.org/wiki/Fichier:Coeloplana_astericola_(16057829918).jpg, GusSar, 2013, https://commons.wikimedia.org/wiki/File:Emu_ (Dromaius_novaehollandiae).jpg, AWS10, Unknown date, https://commons.wikimedia.org/wiki/File:Emu_chick_on_Angas_Downs.jpg, Postdlf, 2010, https:// fr.m.wikipedia.org/wiki/Fichier:Equus_burchelli_at_the_Philadelphia_Zoo_007.jpg, SEFSC Pascagoula Laboratory; Collection of Brandi Noble, NOAA/ NMFS/SEFSC., 2007, https://commons.wikimedia.org/wiki/File:Fish4345_-_Flickr_-_NOAA_Photo_Library.jpg, aquaportail.com, 2007, https://commons. wikimedia.org/wiki/File:Fish4345_-_Flickr_-_NOAA_Photo_Library.jpg, Frank C. Müller, 2007, https://commons.wikimedia.org/wiki/File:Lemniscomys_ barbarus_02_fcm.jpg, Peter Ellis, 2022, https://commons.wikimedia.org/wiki/File:Limax_maximus,_Canberra,_2022-01-07.jpg, Sébastien Vasquez, 2014, https://commons.wikimedia.org/wiki/File:Notocochlis_gualteriana.JPG, Haplochromis, 2009, https://fr.m.wikipedia.org/wiki/Fichier:Ophiolepis_superba_1. JPG, Mokele, 1997, https://qu.wikipedia.org/wiki/Rikcha:Paleosuchus-trigonatus.jpg, Kurt Kulac, 2009, https://commons.wikimedia.org/wiki/File: Pseudopanthera_macularia_kermeter01.jpg, Greg McFall, 2010, https://fr.wikipedia.org/wiki/Fichier:Regal_Sea_Goddess_Nudibranch.jpg, CBG Photography Group, Centre for Biodiversity Genomics, 2016, https://commons.wikimedia.org/wiki/File:Scytodes_intricata.jpg, Orchi, Unknown date, https://fr.m. wikipedia.org/wiki/Fichier:Stanhopea_haseloviana_Orchi_001.jpg, Nick Hobgood, 2005, https://fr.m.wikipedia.org/wiki/Fichier:Synaptula_lamperti_(Sea_ cucumber).jpg, Zac Wolf, 2006, https://fr.m.wikipedia.org/wiki/Fichier:Whale_shark_Georgia_aquarium.jpg, xpda, 2013, https://commons.wikimedia.org/ wiki/File:Xanthippus_corallipes_P1470217a.jpg, Arpingstone, 2005, https://fr.wikipedia.org/wiki/Fichier:Yellow-banded.poison.dart.frog.arp.jpg, Jason Quinn, 2010, https://fr.m.wikipedia.org/wiki/Fichier:Zebra_bullhead_shark_Beijing_Aquarium_17_Sep_2010.jpg, Patrick Giraud, 2006, https://commons.wikimedia. org/wiki/File:Namibie_Etosha_Leopard_01edit.jpg, Yathin S Krishnappa, 2015, https://commons.wikimedia.org/wiki/File:Equus_zebra_hartmannae_-_ Etosha_2015.jpg, Vic Brincat, 2009, https://commons.wikimedia.org/wiki/File:Pomacanthus_imperator_juvenile.jpg, Derek Keats, 2011, https://commons. wikimedia.org/wiki/File:Stapelia_gigantea_(5538239090).jpg, Chiswick Chap, 2012, https://commons.wikimedia.org/wiki/File:Giant_Pufferfish_skin_pattern_ detail.jpg, Holger Krisp, 2013, https://commons.wikimedia.org/wiki/File:Bumblebee_Poison_Frog_Dendrobates_leucomelas.jpg, Chiswick Chap, 2012, https:// commons.wikimedia.org/wiki/File:Giant_Puffer_fish_skin_pattern.JPG, Qwertyz2, 2005, https://commons.wikimedia.org/w/index.php?search= tetraodon&title=Special:MediaSearch&go=Go&type=image, Zac Wolf, 2006, https://commons.wikimedia.org/wiki/File:Whale_shark_Georgia_aquarium.jpg,

Mike Prince, 2014, https://commons.wikimedia.org/wiki/File:Rafflesia_keithii_(13890819315).jpg, Orchi, 2019, https://commons.wikimedia.org/wiki/File:Phalaenopsis_hyroglyphica_Orchi_6752-2.jpg, Christian Kahle, 2021, https://commons.wikimedia.org/wiki/File:Digitalis_purpurea_140940240.jpg, Gunnar Creutz, 2017, https://commons.wikimedia.org/wiki/File:Limax_maximus_in_Floby_1843.jpg, Eric A. Lazo-Wasem, 2019, https://commons.wikimedia.org/wiki/File:Ovalipes_ocellatus_(YPM_IZ_030799).jpeg, Benjamin Bruce, Unknown date, https://commons.wikimedia.org/wiki/File:Dynastes_tityus.JPG, Mnmazur, 2009, https://en.wikipedia.org/wiki/File:Thirteen-lined_ground_squirrel.jpg, Σ64, 2019, https://commons.wikimedia.org/w/index.php?search=scutellaria+rubropunctata&title=Special:MediaSearch&go=Go&type=image, CSIRO National Fish Collection, Unknown date, https://commons.wikimedia.org/wiki/File:Himantura_leoparda_csiro-nfc.jpg, João Carlos Medau, 2013, https://fr.wikipedia.org/wiki/Ocelot#/media/Fichier:Ocelot_(Jaguatirica)_Zoo_Itatiba.jpg, Dysmorodrepanis, 2007, https://commons.wikimedia.org/wiki/File:Ledebouria_socialis2.jpg, Michael Joachim Lucke, 2005, https://commons.wikimedia.org/wiki/File:Aasblume_Aug_2005.jpg, Alain de Trenqualye, 2010, https://commons.wikimedia.org/wiki/File:Alstroemeria_garaventae.jpg, Michael Lejeune, 2007, https://fr.m.wikipedia.org/wiki/Fichier:Calceolaria_uniflora.jpg, Evelyn Simak, 2008, https://commons.wikimedia.org/wiki/File:Common_foxglove_(Digitalis_purpurea)_-_detail_of_flower_-_geograph.org.uk_-_855282.jpg, Mario Vallejo-Marin, Ph.D., 2012, https://en.wikipedia.org/wiki/File:Erythranthe_peregrina.jpg, Hectonichus, 2013, https://commons.wikimedia.org/wiki/File:Saxifragaceae_-_Saxifraga_rotundifolia.jpg, 阿橋 HQ, 2004, https://commons.wikimedia.org/wiki/File:%E7%B7%9E%E8%8A%B1%E8%94%93_Alsobia_Cygnet_-%E9%A6%99%E6%B8%AF%E5%98%89%E9%81%93%E7%90%86%E8%BE%B2%E5%A0%B4_Kadoorie_Farm,_Hong_Kong-_(9237449975).jpg.

species) or PGTCPs (73 reliable species, 79 potential). Groups with relatively few species such as Feliformia (to which the leopard belongs) have a large proportion of species bearing PGTCPs, while others with a high specific richness exhibit few or no PGTCPs, such as the bats (Chiroptera).

**Table 1. Potential causes for Turing-like patterns and PGTCPs sparsity in specific taxa.**

| Taxon | Classic Turing-like Colour patterns estimated sparsity (+, ++, +++, absence) | PGTCPs estimated sparsity (+, ++, +++) | Potential causes for Turing-like patterns (*in italics*) and PGTCPs (**in bold**) sparsity, other than chance. |
|---|---|---|---|
| Insecta | ++ | +++ | • *Segmented and compartmented body (too little space to produce proper motifs)*<br>• *Segmented and compartmented body (the pre-existing repeated structures of the body may favor other types of colour pattern mechanisms, as the ability of Turing systems to produce repeated patterns may be de facto obsolete)*<br>• **Growth modalities of insect body parts often involve a discontinuous growth and a small absolute increase in tissue surface at each iteration, making it potentially more difficult to form PGTCPs.** |
| Aves | Juveniles (+), adults (+++) | Juveniles (+++), adults (+++) | • *Many juveniles have no down nor colored skin. Disordered down or a too low feather density could hide potential Turing-patterns or PGTCPs (by decoupling or pixelation).*<br>• *The feathers of adults make it difficult or impossible for a skin-level 2D Turing pattern to keep its visible geometry at the scale of the plumage (decoupling).*<br>• **Many species harbor repeated patterns on their feathers, but the underlying mechanism is** *a priori* **not compatible with the 2D growth necessary for PGTCPs formation.** |
| Fungi | Absent? | Absent? | • *Physical cracking patterning (such as the very famous* Amanita muscaria *red and white pattern or black and white desquamation pattern of* Coprinopsis picacea*) could have been preferentially selected during Evolution at the expense of potential Turing mechanism).*<br>• *Ecologically, there might by a reduced evolutionary pressure to produce repeated colour patterns on Fungi caps (even if Amanitaceae produce some striking ones) and other ways to attract/repel other species might have been favored like poisons or attractive flavors. Many Fungi taxa seem to exhibit a poor relationship between colors and attractiveness/repellency.*<br>• *Turing patterns not involved in coloration might be responsible for the great hymenium shape diversity* [53]. **This putative ability of Fungi to produced Turing patterns, together with their absence for colour patterning strengthens the previous point.** |
| Angiosperms | ++ | + | • *Families whose species are not based on zoogamy seem to be spared from repeated colour patterns, which would then explain the absence of classic Turing patterns and PGTCPs.*<br>• **Some species could produce prepatterns of PGTCPs but may be hidden when the colors are produced, by blurring. We have observed many orchid species that keep PGTCPs stigmas (like spots with tortuous edges) but many of the typical characters are absent (rosettes, intermediate colors).** |

In orchids, where only PGTCPs have been searched for, 525 species (1.8%) clearly display PGTCPs, 528 others are potential candidates. Again, while many genera possess at least one species with a PGTCP, some contribute more both in proportion and in absolute value, such as the genus *Stanhopea*, of which 33 of the 81 species (41%) display conspicuous PGTCPs.

This first overview of PGTCPs in Eukaryotes suggests a very high diversity, which deserves to be the subject of a greater attention in the future and raises the question of the selective forces that may have driven the evolution of PGTCPs.

## Disruption induced by tissue growth can generate new shapes like rosettes, lines-and-dots alternation or intermediate bands, together with new colors by mixing and memory mechanisms in Turing patterns

The iconic pattern of the leopard has been the focus of much attention over the last twenty years, where several mechanisms of rosette formation have been proposed *(15, 16)*. The current consensus is that a rosette is an old dot reorganized into smaller dots in response to the increase in tissue size (Fig 2A and 2B). Because Turing systems are sensitive to scale, an extended motif driven by tissue growth will be out of its equilibrium and cause its rearrangement. PGTCPs are then the more or less transient states produced during a return toward the equilibrium. The second particularity of leopard rosette, *i.e.*, the presence of orange coloration in its center, intermediate between dark brown and light yellow background, has been recently reproduced by de Gomensoro Malheiros et al. [20] by growth combined to a memory mechanism where each particular location of the final pattern is the average of the all the previous states throughout the pattern formation. We extended this model to the emu and reproduced its intermediate band pattern (Fig 2C). One of the biological explanations to the leopard rosettes and the emu intermediary bands could be that melanocytes are regularly produced in the skin tissues, becoming black (eumelanin) or yellow (pheomelanin) melanocytes depending on the current local state of Turing system, and at the end, tissues regions where Turing state knew the two states through time are composed of a mosaic of black and yellow melanocytes. This leads to intermediary color at the macroscopic scale (by our eye interpretation), in this case the orangish hair of leopard and light-brown feathers of the emu.

As we saw, these patterns combine both novel shapes and colors, two possible effects of growth. Alternatively, some species can exhibit shape effects without intermediary color, like the orchid *P. hieroglyphica*, which on the other hand has stretched rosettes. We reproduced them by simulated anisotropic growth without memory (Fig 2A, up and 2B, right).

To better classify growth effects and PGTCPs, we suggest that rosettes belong to a bigger family of disrupted patterns. Considering the three classic Turing motifs shapes (dots, stripes and mazes), rosettes are then disrupted spots, intermediate bands and stripes-and-spots alternations are disrupted stripes, and nested patterns of some pufferfishes like *Tetraodon mbu* are mazes disrupted into mazes of mazes [25]. By varying Turing parameters in our growth model, we reproduced the latter (Fig 2F).

Despite the efficiency of this memory model, it is *a priori* not compatible with all biological situations including some systems now well described, such as the zebrafish (*Danio rerio*) stripes formation. Indeed, although we observe intermediate colors in a large number of Actinopterygii fish species, the cellular Turing mechanisms described in zebrafishes are not compatible with the conception of additive coloration through memory of earlier stages proposed for the leopard, as these fish colored cells constantly rearrange [10, 15]. We therefore propose a new type of mechanism that could fit the current biological knowledge of zebrafishes stripe formation, without superimposition of several color states during tissue growth, which we called "non-equilibrium intermediate coloration".

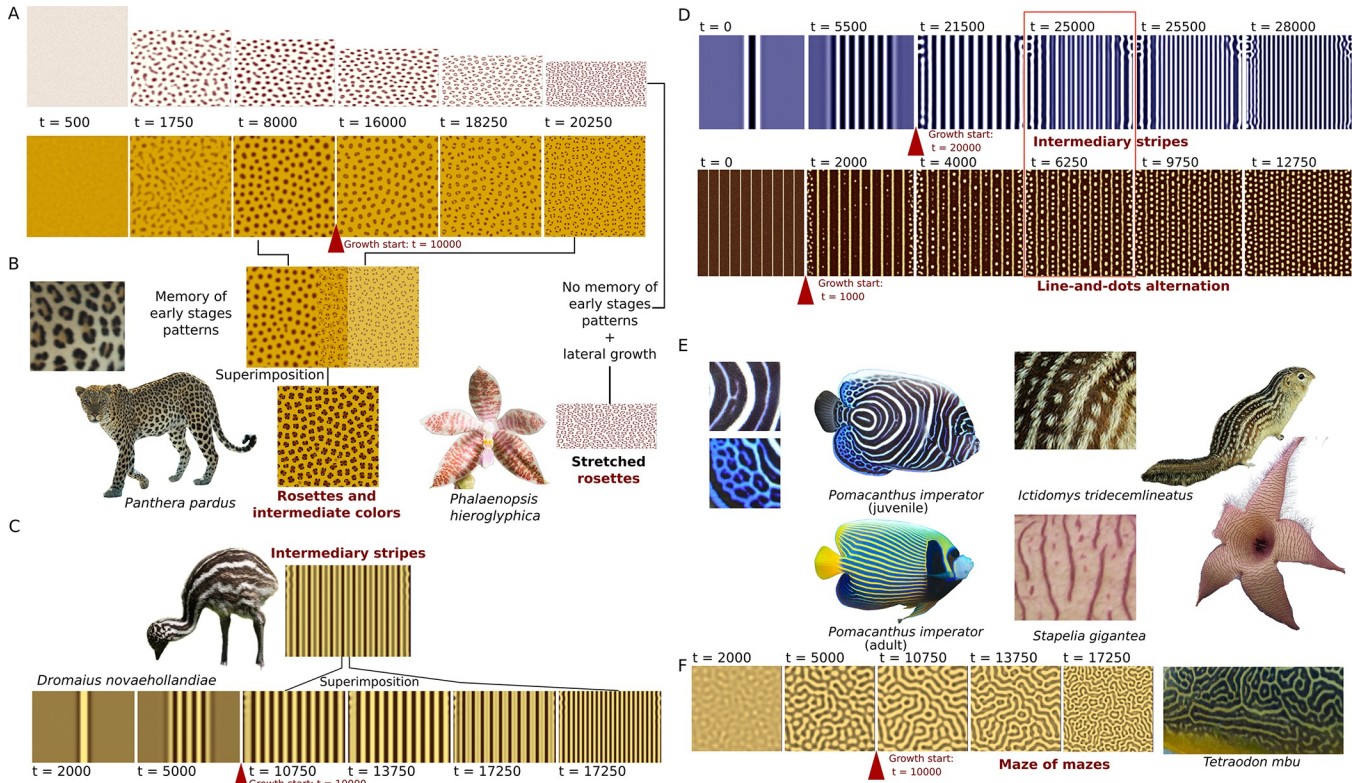

**Fig 2. Modelling of tissue growth reproduce all types of PGTCPs via diverse mechanisms.** (**A**) First row: simulation of *Phalaenopsis hieroglyphica* rosette patterns (with higher lateral growth). Second row: simulation of leopard (*Panthera pardus*, picture under a CC BY license, with permission from Patrick Giraud, original copyright 2006, https://fr.wikipedia.org/wiki/L%C3%A9opard#/media/Fichier:Namibie_Etosha_Leopard_01edit.jpg) rosette patterns (with uniform growth). (**B**)-(**C**) Leopard (rosettes) and Emu (intermediary bands) patterns are obtained through superimposition of early and late patterns via a memory mechanism, absent in *P. hieroglyphica* (picture under a CC BY license, with permission from Motohiro Sunouchi, original copyright 2020, https://commons.wikimedia.org/wiki/File:Phalaenopsis_hieroglyphica_%27Mindanao_-1901%27_%28Rchb.f.%29_H.R.Sweet,_Amer._Orchid_Soc._Bull._38_36_%281969%29_%2849661185378%29.jpg). (**D**) First row: simulation of *Pomacanthus imperator* intermediate band pattern, by out-of-equilibrium mechanism. Second row: simulation of line-and-dots patterns, with the example of *Ictidomys tridecemlineatus*. (**E**) Overview and details of *P. imperator*, *I. tridecemlineatus* and *Stapelia gigantea* PGTCPs. (*P. imperator* juvenile picture under a CC BY license, with permission from Vic Brincat, original copyright 2009, https://commons.wikimedia.org/wiki/File:Pomacanthus_imperator_juvenile.jpg), (*I. tridecemlineatus* picture under a CC BY license, with permission from Mnmazur, original copyright 2009, https://en.wikipedia.org/wiki/File:Thirteen-lined_ground_squirrel.jpg?uselang=en#Licensing) (*S. gigantea* picture under a CC BY license, with permission from Michael Joachim Lucke, original copyright 2005, https://commons.wikimedia.org/wiki/File:Aasblume_Aug_2005.jpg (**F**) Simulation of maze of mazes pattern, with the detail of *Tetraodon mbu* adult skin pattern (picture under a CC BY license, with permission from Chiswick Chap, original copyright 2012, https://commons.wikimedia.org/wiki/File:Giant_Puffer_fish_skin_pattern.JPG).

In our hypothesis, the intermediate colors are produced by intermediate values of the Turing system parameters (typically activator or inhibitor medium-level concentrations). When tissue growth disrupts an already formed pattern, for example stripes, new ones are generated from the middle of the pre-existing ones. At the equilibrium, new sharp frontiers will be generated, but during their formation, intermediate values may be reached over a sufficiently long period and over a large width to be observed, (Fig 2D, 1st row). This implies that if colors are produced in real time, species will display intermediate coloration during their growth phase and intermediate color will be transiently visible (e.g., *Pomacanthus imperator* juvenile pattern, Fig 2E). Furthermore, if the pattern dynamics are stopped before reaching its equilibrium, intermediary colors can become definitive by being snap-shot. Indeed, many natural PGTCPs could be 'out of equilibrium' patterns frozen in time by the biological readout of the Turing system state at a given time.

This doublet of mechanisms may also explain why intermediate colors are rare in Angiosperms (even though rosettes are very common, showing that growth can effectively disrupt many plant color patterns). According to the pioneer cellular and genetic study by Ding *et al.* [16] on the flowers of *Mimulus*, their color mechanism is based on a reaction-diffusion involving two proteins. The preservation of memory seems more complicated (although not impossible) to imagine this at the molecular level [26], given the short half-life of molecules. Nevertheless, some flowers like those of the Lamiaceae *Scutellaria rubropunctata* exhibit striking leopard-like purple rosettes, with intermediary colors on their middle, showing that is it not totally impossible to produce them in Angiosperms [27].

The last geometrical category of PGTCPs, lines-and-dots alternation is a bit different in terms of mechanics.

Less common than other PGTCPs, it is harbored by very distantly-related species like *Stapelia gigantea* on its flowers, the squirrel *Ictidomys tridecemlineatus* on its fur (Fig 2E) or the whale shark *Rhincodon typus* on its skin. These species generate dots or sometimes mazes in the middle of bands. Several models have been proposed to reproduce them, some related to biological phenomena such as neural crest migration [20]. Our proposition is based on both internal and external factors, respectively random noise of the system and accelerating growth of the tissue.

A stripe as initial condition (1st generation stripe, Fig 2D, 2nd row, t = 0 and t = 1000) generates a multi-striped pattern (2nd generation stripes, Fig 2D, 2nd row, t = 5000 and t = 22000) in a small tissue. If a higher growth rate follows, eventually helped by a highly noisy Turing system, this could disrupt the formation of 3rd generation stripes that will break into aligned dots (Fig 2D, 2nd row, t = 22700), leading to alternations of stripes and dots. We can note that like all PGTCPs, these patterns are only meso-stable, if the dynamics of the system continues to be active, the fate of all stripes is to break into dots (Fig 2D, 2nd row, t = 10000, biologically observe in whale sharks) unless a forcing mechanism maintains them or if the pattern is snapshot. Lines-and-dots patterns are a family containing variations, if growth is very high, more complex patterns than aligned dots can form [20].

In all of our simulations, in order to simplify, we chose to let the Turing-like system to start in a non-growing domain, but it is not a mandatory condition. Then, at one point, growth starts, and we chose to make growth rate constant so the tissue surface grows exponentially, which is classical during development.

In summary, all the types of PGTCPs (rosettes, intermediate colorations, line-and-dots alternation, intermediate bands and all their variations) could be view as the result of a triad: i) tissue growth, ii) scale sensibility of Turing-like systems and iii) system memory, as Turing systems dynamically rearrange new patterns from the previous ones as new initial conditions.

While a definitive functional proof of the implication of growth is still lacking, we then gather a cluster of clues suggesting that it is the main candidate for the great diversity of PGTCPs in Eukaryotes.

## The distribution of PGTCPs occurrences suggests a universal mechanism, and morphometric analyses show that their presence correlates with the most-growing areas of the tissues

The putative parameter(s) conditioning the presence or absence of PGTCPs are undoubtedly common and universal, as evidenced by the examples of presence/absence found at all scales of life starting with the individual scale, in a spatial and temporal manner (Fig 3A). Indeed, in most species with PGTCPs on their external tissues, we noticed that not all the tissue is covered by PGTCPs. At the temporal scale, the fish *Pomacanthus imperator* has only intermediate

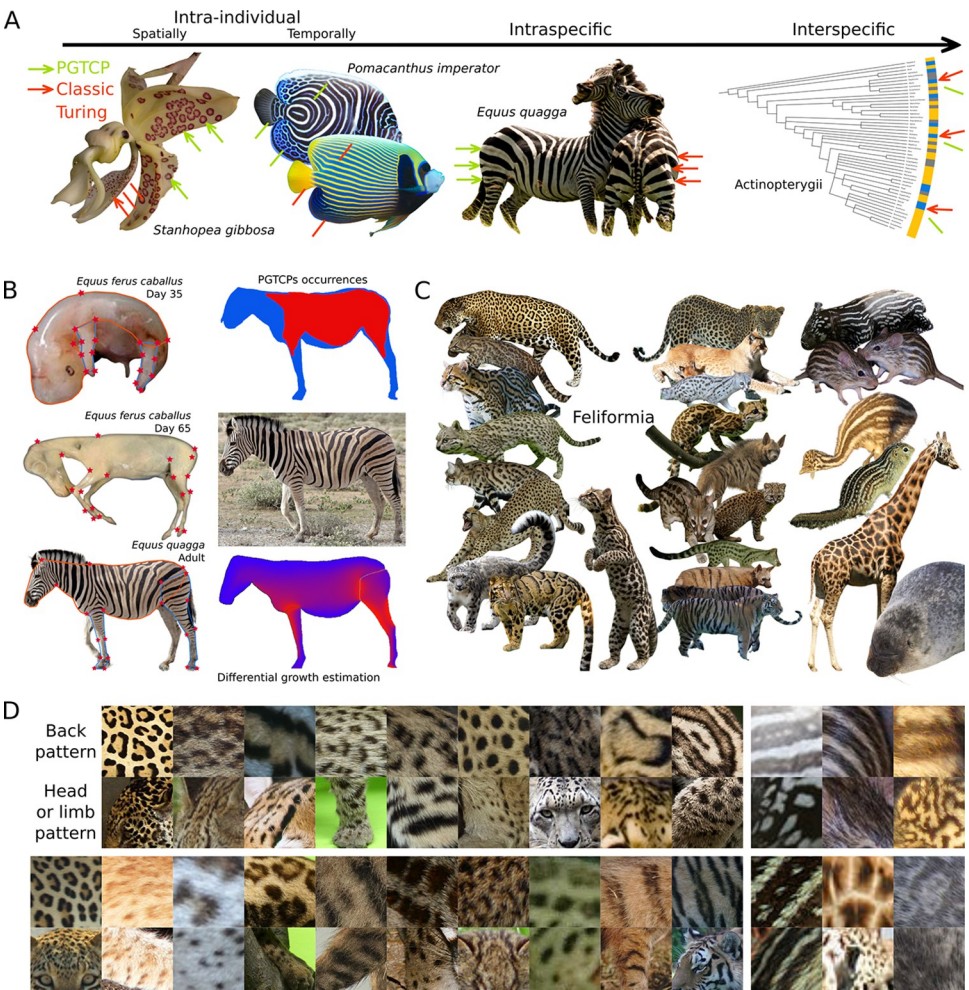

**Fig 3. PGTCPs correlate with most-expensing regions of the tissue, suggesting the responsibility of growth in their formation.** (**A**) Examples of presence/absence of PGTCPs are found at every biological scale, suggesting a universal mechanism of control, growth being the main candidate (*P. imperator* juvenile picture under a CC BY license, with permission from Vic Brincat, original copyright 2009, https://commons.wikimedia.org/wiki/File: Pomacanthus_imperator_juvenile.jpg). (**B**) Approximation of zebra (*Equus quagga*) skin growth via morphometrics during development (in horse embryo, pictures reprinted from [29] under a CC BY license, with permission from Maria Angelica Miglino, original copyright 2016, and in zebra adult, left panel). Stars represent the homologous landmarks, used in geometric morphometric studies, lines are the homologous outlines of the organism bodies. Correlation between predicted most-expensing regions (up), and PGTCPs occurrences (down), right panel. (**C**)-(**D**) Examples of occurrences of PGTCPs restricted to the back (high-growing region), while only classic Turing patterns present on the head or the limb (low-growing regions) in nineteen Feliformia, six other mammal species and Emu juvenile (*Dromaius novaehollandiae*). Pictures sources for C and D panels: Charles J. Sharp, 2015, https://commons. wikimedia.org/wiki/File:Jaguar_(Panthera_onca_palustris)_male_Three_Brothers_River_(cropped)_cropped.jpg, João Carlos Medau, 2013, https://fr.wikipedia.org/wiki/Ocelot#/media/Fichier:Ocelot_(Jaguatirica)_Zoo_Itatiba.jpg, Thomas Fuhrmann 2015, https://commons.wikimedia.org/wiki/File:Masai_Mara_National_Reserve_07_-_cheetah_% 28Acinonyx_jubatus%29.jpg, Marit & Toomas Hinnosaar, 2009, https://commons.wikimedia.org/wiki/File:Uncia_ uncia_-_A_tail_and_a_cat_cropped.jpg,kellinahandbasket, 2007, https://fr.wikipedia.org/wiki/Panth%C3%A8re_n% C3%A9buleuse#/media/Fichier:Clouded_Leopard_SanDiegoZoo.jpg, Charles J. Sharp, 2018, https://fr.m.wikipedia. org/wiki/Fichier:Spotted_hyena_(Crocuta_crocuta).jpg, Ltshears, 2011, https://fr.m.wikipedia.org/wiki/Fichier: BlackFootedCat57.jpg, Altaileopard, 2006, https://commons.wikimedia.org/wiki/File:Genetta_tigrina.jpg, Gecko-kus, 2014, https://fr.m.wikipedia.org/wiki/Fichier:Pardine_Genet_at_WWP.jpg, Bernard DUPONT, 2015, https://fr. wikipedia.org/wiki/Genetta_tigrina#/media/Fichier:Large-spotted_Genet_(Genetta_tigrina)_(17182311499).jpg, Rushikesh Deshmukh DOP, 2019, https://commons.wikimedia.org/wiki/File:Striped_hyena_(Hyaena_hyaena) _-_cropped.jpg?uselang=fr, Mauro Tammone, 2019, https://commons.wikimedia.org/wiki/File:Striped_hyena_ (Hyaena_hyaena)_-_cropped.jpg?uselang=fr, Daderot, 2013, https://fr.wikipedia.org/wiki/Leopardus_guttulus#/ media/Fichier:Leopardus_tigrinus_(Felis_tigrina)_-_Museo_Civico_di_Storia_Naturale_Giacomo_Doria_-_Genoa, _Italy_-_DSC02677.JPG, Groumfy69, 2014, https://fr.m.wikipedia.org/wiki/Fichier:Leopardus_tigrinus_-_Parc_des_F

%C3%A9lins.jpg, Supreet Sahoo, 2016, https://fr.m.wikipedia.org/wiki/Fichier:Margay_in_Costa_Rica.jpg, Hafiz Issadeen, 2016, https://fr.m.wikipedia.org/wiki/Fichier:Margay_in_Costa_Rica.jpg, Patrick Giraud, 2006, https://fr.wikipedia.org/wiki/L%C3%A9opard_d%27Afrique#/media/Fichier:Namibie_Etosha_Leopard_01edit.jpg, Abujoy, 2017, https://fr.m.wikipedia.org/wiki/Fichier:La_Bourbansais_10_Tigres_de_Sib%C3%A9rie.jpg, Derek Keats, 2019, https://fr.m.wikipedia.org/wiki/Fichier:Aardwolf,_Proteles_cristata,_at_Lion_and_Rhino_Reserve,_Gauteng,_South_Africa,_(47987215058).jpg, AWS10, Unknown date, https://commons.wikimedia.org/wiki/File:Emu_chick_on_Angas_Downs.jpg, Adrian Pingstone, 2003, https://fr.m.wikipedia.org/wiki/Fichier:Giraffa_camelopardalis_rothschildi.jpg, Mnmazur, 2009, https://en.wikipedia.org/wiki/Thirteen-lined_ground_squirrel#/media/File:Thirteen-lined_ground_squirrel.jpg, Azay 2007, https://fr.wikipedia.org/wiki/Lemniscomys_barbarus#/media/Fichier:Lemniscomys_barbarus01.JPG, Lee Cooper, 2001, https://fr.m.wikipedia.org/wiki/Fichier:Ringedsealportrait.jpg, Sasha Kopf, 2007, https://commons.wikimedia.org/wiki/File:Princess_Tapir_sleeping.JPG.

bands during the transition from juvenile to adult morphs (Fig 3A). At the intraspecific scale, we documented individuals belonging to the same species, some of them displaying these patterns and others not, for instance in zebras (*Equus quagga*, Fig 3A). Finally, the large distribution in the phylogenetic tree of these motifs presented in Fig 1B and 1E shows that if a common mechanism governs all these motifs, it must be shared by a large amount of very diverse species. All these examples suggest the existence of a widespread mechanism underlying the diversity of patterns, and growth fits this requirement.

To strongly suggest its role in PGTCPs formation, we built an algorithm based on morphometric measurements (Fig 3B, left) to estimate the tissue areas undergoing the greatest cumulative expansion during the development of organisms whose developmental sequence was available (Fig 3B, upright) and compared them to the map of presence or absence of growth patterns (Fig 3B, downright). In the Burchell's zebra (*Equus quagga burchellii)*, areas where growth is predicted to be more intense correlate rather satisfactorily with the areas of presence of intermediate bands, *i.e.*, on the trunk and on the proximal part of the limbs, with an absence of patterns on the head, neck and distal part of the limbs. It is also supported by the fact that in most mammals, PGTCPs are preferentially located on the trunk and absent from the head, which is in agreement with the fact that the head undergoes much less growth than the body during the embryonic development [28]. Fig 3C and 3D. show a representative panel of Feliformia species (plus six species of other mammals and one bird) exhibiting PGTCPs only on their back, while their head or limb present only classic Turing patterns.

Together, these indirect clues strongly suggest that growth is the main responsible for PGTCP formation in Eukaryotes.

## Effects of growth on Turing patterns can be used as reverse tools to extract information on pattern timing and system characteristics such as wavelength

The effects of growth on Turing patterns reported in this article are reminiscent of earlier stages: rosettes are ancient dots, mixed colors are produced in ancient dark or light zone, and so on.

This led us to ask if this could be exploited to obtain information on earlier stages, in particular the crucial timing of pattern formation, which is invisible to the experimenter in many species, because very often production of colors occurs much later [30]. Such data could increase both the raw knowledge of Turing pattern systems together with allowing comparative analyses, but also it could facilitate the implementation of functional experiments by allowing to focus on a reduced temporal window.

Periodicity is the crucial characteristic of Turing systems for the reverse method we have developed (presented in S1A Fig). Indeed, by assuming that for a given organism i) the whole pattern is set up at the same time in all points of the tissue and that ii) the parameters of the

system are homogeneous in space and time, we could deduce that at the time of the formation of the pattern, all the motifs must be regularly spaced on the tissue and have the same size. However, since for a large part of the species this is not what is observed on the final pattern, the cause of this discrepancy is the differences in the relative growth between the tissue parts during development, from the stage of the end of pattern production to the moment of observation.

By extension, concerning growth patterns such as rosettes, if we consider either a rosette as a single motif, or several motifs (by counting how many sub-motifs it is divided into), it could theoretically be possible to calculate both the timing of the initial dots pattern, and the pattern of rosette formation.

To test this hypothesis, we use the cat (*Felis silvestris*) and leopard models, the first having a well described embryogenesis for which morphometric measurements are applicable, the second for its PGTCPs, while being morphologically close to the first at least until its juvenile stages (S1B Fig). Furthermore, when we look at the estimates of the 2nd generation patterns, it points to much more recent stages (S1B Fig), juvenile or even adult, which is coherent with the observations of Liu et al. [19], which show that the pattern continues to be set up after the juvenile stages.

To improve predictions, we simulated the formation of rosette patterns at different growth rates (S1C Fig, solid lines), and compared to the theoretical number of motifs at each step (S1C Fig, dashed lines), i.e., at an equivalent size, but without growth.

This showed that the relationship between growth rate and motif number is not totally trivial and must be used with great caution in our predictions.

Finally, we develop a method to better follow and estimate tissue growth thanks to artificial landmarks. Some plant tissues like sepal outer epidermis are accessible during their development and therefore available for marking. We chose to simply apply spot landmarks with waterproof ink. This relatively non-invasive technique allows to estimate tissue deformations and so local growth through developmental time. By tracking the relative position of ink points using 2D photography, or *in vivo* photogrammetry (S1D Fig, lower panel) in the case of 3-dimensional folded tissues, geometric morphometry techniques can be applied.

Although imperfect, these methods should be able to estimate the timing of production of classical Turing and growth patterns on many species and allow large-scale comparative studies (S1D Fig). Moreover, as precise patterning timings will be revealed by functional experiments, the validity of the prediction of this method could be verified and improved. Finally, while the proxy used here is the number of motifs, a study assisted by deep learning could help find finer estimators [31], which would therefore improve the prediction of the timing, and its corollary: the natural wavelength of the pattern.

## PGTCPs could have evolved under selection for enhancing or reducing pattern visibility

Resulting from the interaction between a Turing system and tissue growth contingent to the development, PGTCPs could appear as only by-products of growth and not under the scope of an efficient selection. However, we have previously documented numerous examples of presence/absence of growth patterns, within and among species, suggesting instead possible ways of control and selection.

Like color patterns in general, classic Turing-like color patterns are perceived by con- and hetero-specifics, and could therefore be under selection for signaling or camouflage [32]. We therefore wondered whether the geometric and color features of the PGTCPs could improve signaling or camouflage compared to classical Turing patterns.

Concerning camouflage, the mixed colorations caused by growth could allow the species displaying them to retain the camouflage capacities of Turing patterns repetition, while reducing the segregation of colors that is intrinsically linked to them. Indeed, a classic Turing system produces a segregation of colors, not very compatible with the production of intermediate colors, which are likely to be useful in camouflage. For example, the shades of brown resulting from the mixture of pigment cells in the eumelanin (black) / pheomelanin (yellow/orange) system are the basis of many camouflages, especially in mammals and birds [33, 34]. If dark and light melanocytes are separated by the Turing system, the resulting patterns are often very contrasting, as they lose the potential for color shades. This is illustrated by the many examples of mammalian species with black and light Turing patterns from a genus where their closely related species have intermediate-colored coats (Fig 4A). By allowing the displaying of intermediate colors, the effects of growth could then be beneficial in terms of camouflage and be

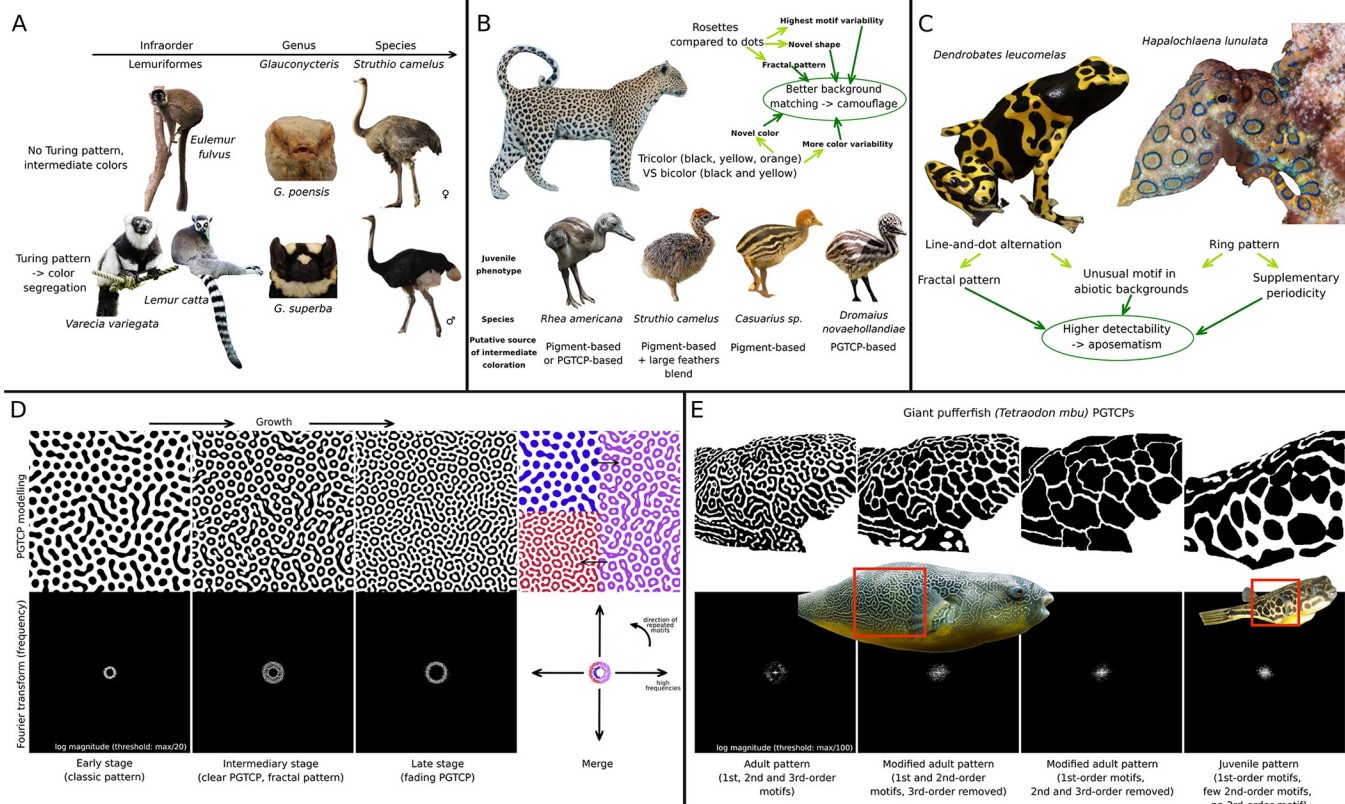

**Fig 4. Potential biological functions enhanced by PGTCPs.** (**A**) Turing systems are a source of colour segregation, leading to contrasted patterns with more extreme and less intermediate colorations. Species or individual with intermediary colour patterns (up) compared to relative ones with Turing patterns (down). (**B**) Intermediate colors and novel shapes induced by growth could improve camouflage of species. Up: leopard rosettes have characteristics that can potentially improve the effectiveness of camouflage. Down: Different ways of producing intermediate colors, improving camouflage in juvenile Ratites. (**C**) *Dendrobates leucomelas* line-and-dots alternations and *Hapalochlaena lunulata* ringed rosettes have characteristics that can potentially improve the effectiveness of their aposematic pattern, like unusual motifs, more fractal and multiple periodic patterns. (**D**) Discrete Fourier Transform (DFT) of ring pattern simulation show that PGTCPs (2nd panel) present both the spectrum characteristics of early stage (1st panel) and late stage (3rd panel), marking their multi-periodicity pattern. (**D**) DFT of *Tetraodon mbu* maze of mazes PGTCP shows that adult pattern (with 1st, 2nd and 3rd-order motifs, 1st panel) presents a more complete spectrum than juvenile pattern (4th panel, only 1st order motifs) and simplified patterns (2nd and 3rd panels). Pictures sources: prilfish 2009 https://commons.wikimedia.org/wiki/File:Chelidonura_livida,_Dahab.jpg, Charles J. Sharp, 2019 https://en.wikipedia.org/wiki/Common_brown_lemur#/media/File:Common_brown_lemur_(Eulemur_fulvus)_male.jpg, Mathias Appel, 2015, https://commons.wikimedia.org/wiki/File:Black_and_white_Ruffed_Lemur_(22383460999).jpg, Architas, 2018, https://commons.wikimedia.org/wiki/File:Lemur_catta_2018.jpg, Reeder D, Helgen K, Vodzak M, Lunde D, Ejotre I, 2013, https://commons.wikimedia.org/wiki/File:Contrasting_facial_aspects_for_Glauconycteris_cf._poensis_and_Niumbaha_superba_-_ZooKeys-285-089-g003.jpeg, H. Zell, 2018, https://commons.wikimedia.org/wiki/File:Struthio_camelus_-_juvenile_-_R%C3%BClzheim.jpg, Qwertzy2, 2005, https://commons.wikimedia.org/w/index.php?search=tetraodon&title=Special:MediaSearch&go=Go&type=image, GusSar, 2013, https://commons.wikimedia.org/wiki/File:Emu_(Dromaius_novaehollandiae).jpg, Rickard Zerpe, 2019, https://commons.wikimedia.org/wiki/File:Greater_blue-ringed_octopus_(Hapalochlaena_lunulata)_(48272090161).jpg, Holger Krisp, 2013, https://commons.wikimedia.org/wiki/File:Bumblebee_Poison_Frog_Dendrobates_leucomelas.jpg.

positively selected during evolution. For example, the geometrical particularities of some growth patterns such as leopard rosettes could further improve the camouflage efficiency compared to a classical spotted pattern, by adding highest motif variability, novel shapes and a more fractal pattern (Fig 4B, [35, 36]). We also observe that all the Ratites juveniles possess orange and brownish colors but for each of these four species, the origin and nature of these intermediate colors seem different: Rhea and emu (*Dromaius novaehollandiae*) exhibit orange intermediate bands, i.e., PGTCPs (Fig 4B, down left and right), the cassowary *(Cassuarius sp.)* has orange colors on its head and neck, probably due to a particular type of pheomelanin [37] as for the juvenile ostrich (*Struthio camelus*) which also has very large feathers (for a down coat) intermingled on its back, leading to the effect of averaging and mixing the colors produced by the melanocytes in the skin. These convergent traits could be a sign that intermediate colors are under selection and have a specific biological function. In particular, we can hypothesize that they enhance their cryptic pattern, making it more difficult to detect by potential predators (Fig 4B, upright) [38]. Growth would thus be a mean to produce intermediate colors.

On the other side of the spectrum, some characteristics of growth patterns could enhance their visibility and their attractiveness or repulsion to conspecifics or other species, as in the case of aposematism. Indeed, the periodicity of Turing patterns makes them easily detectable by a large number of visual neural networks, and their attractiveness, including in the human species, is directly related to their low processing time for the brain [39]. We hypothesize that growth could be a non-negligible source of striking geometries, positively selected during evolution. Typically, the aposematic ring patterns of the greater blue-ringed octopus (*Hapalochlaena lunulata* [40]) and Dendrobatid frogs [41, 42] line-and-dots patterns have all the characteristics to have been produced by growth (Fig 4C) and it can be hypothesized that these motifs improve pattern detection, compared to classical dotted patterns. Contrary to periodic stripes or dots [43], ring patterns and line-and-dot alternations are at our knowledge very rare or absent in abiotic environments, which might confer them a higher detectability by vision systems. Furthermore, by exhibiting supplemental periodicities than classic Turing patterns, we hypothesize that they could be better and more detectable by con- and hetero-specifics. To quantify this potential add value of PGTCPs, we performed Fourier analysis on simulated and biological patterns (Fig 4D and 4E).

We reproduced ring formation, and the frequency magnitude spectrum of the different states showed that the intermediate state which present a clear PGTCP (Fig 4D) combines the periodicity characteristics of both the initial state (classic Turing spot pattern) and the late stage (return to a classic Turing spot pattern). Furthermore, we obtain the same trend in the patterns of leopard and blue-ringed octopus (S2B and S2C Fig) and for the maze of mazes pattern of *Tetraodon mbu* (Fig 4E). The adult pattern -which comprise 3 different scales of motifs- exhibits a more complete spectrum than both the juvenile large spot pattern and the modified patterns with $2^{nd}$ and/$3^{rd}$ order motifs removed, marking the presence of more periodicities.

More than mere by-products of developing tissue growth, we showed that growth patterns could potentially enhance camouflage capabilities, and conversely improve pattern visibility in situations of aposematism, interspecific attraction or sexual selection. Behavioral and evolutionary studies would then provide an even more complete picture of the functional possibilities of these particular patterns.

## Discussion

### Factors controlling the taxa-specific absence or sparsity of putative-growth Turing color patterns (PGTCPs)

Although PGTCPs are widely distributed in the phylogeny tree of Eukaryotes, there are hotspots (e.g., Chondrichthyes and Actinopterygii for Animals or Lamiids for Angiosperms) and

quasi-empty areas (e.g., Rosids for Angiosperms or Fungi (tree not shown)). As PGTCPs are the combination of growth and an active Turing color system, the absence of the latter is a first trivial reason of the absence of PGTCPs, and it can be spot in the phylogeny in the grey zones (neither blue nor yellow in Fig 1). In contrast, there are some taxa with some or many examples of species with classic periodic color patterns, but in which it was impossible or very difficult to find examples of PGTCPs, typically birds where to our present knowledge, only one species exhibits a PGTCP, the emu *Dromaius novaehollandiae*. As the explanations could be very different from one taxon to another, we tried to propose several hypotheses, which are summarized in Table 1.

We then detail here one example, the case of insects, perhaps the most informative in many ways for the factors controlling the presence/absence of PGTCPs. Indeed, insects are the metazoan class containing the largest number of species (estimated at more than 5 million species [44]) and are extremely diverse in terms of color patterns, both for colors and shapes. We would expect to find many examples of classical Turing patterns, and also of PGTCPs, which is not the case for either. More surprisingly, many closely related taxa belonging like Insects to the Pancrustaceae, exhibit numerous Turing color patterns and PGTCPs, Decapoda being one example. Last but not least, Turing systems have been found in Insects, but producing other types of structures than color patterns (such as Turing nanopatterns of coat corneae [45] or tarsal attachment patterns [46]), showing the compatibility between the external tissues of insects and Turing systems.

So, to explain this apparently rarity of Turing color patterns and PGTCPs in insects, we invoked three types of explanations, not mutually exclusive. First, many parts of insect bodies are segmented and compartmented, such as the abdomen or wings. This causes a division of space and could reduce the size of the tissue where a hypothetical Turing system must express. The boundaries between compartments may be too disruptive for Turing system actors and prevent them from forming a large-scale proper pattern. If Turing systems could theoretically be produced at any scale, as evidenced by the nanopatterns mentioned above, this compartmentalization may prevent large motifs to occur, and therefore causing an impossibility to ensure a proper biological function (*e.g.*, too small and tight motifs may not be visible to inter or intra-specific individuals). The fact that insects are on average small organisms goes in the same direction, as larger animals would be favored for the formation of proper Turing color patterns. This is supported by the fact that the few examples of PGTCPs that we found in insects were in relatively large body parts and relatively large species, typically wings of *Pseudopanthera macularia* (Lepidoptera), *Eupteryx decemnotata* (Hemiptera) *Xanthippus corallipes* (Orthoptera) and *Dynastes tityus* (Coleoptera). Second, because insect body is already segmented, it may favor other types of color pattern mechanisms and thus disfavor Turing patterning during evolution (for example, black and white stripes in some silkworm strains are simply generated by the expression of Toll ligand Spätzle3 gene in the anterior part of every segment [47], generating a bicolor pattern, but *de facto* leads to a multi-repeated pattern thanks to the presence of multiple body segments). Finally, the growth conditions of insect outer tissues may also be a hindrance in the formation of PGTCPs, as most outer tissues are renewed by molting [48]. This discontinuous growth may favor i) equilibrium patterns that disfavor PGTCPs, ii) the resetting of the Turing system at each molt and the *de novo* production of a color pattern, with a little increase in tissue absolute size because its step-by-step nature, thus again disfavoring PGTCPs. To summarize, the optimal conditions for observing PGTCPs are a tissue which a large increase in its absolute size and a continuous growth, which is more or less the opposite of the insect growth modalities. We believe that this reasoning can be extended to other taxa and be the subject of further and more precise studies.

## Hidden PGTCPs: Spatial decoupling, blurring, or pixelating

From the Turing-like system to the final coloration phenotype many mechanisms may act as filters or transformers, also called "secondary mechanisms" [24] taking a rough molecular state (in case of classic Turing based on morphogens) to a complex color pattern, the one which is observed in the end. Information and complexity can be added through the process, or removed, or even translated into another nature. For this reason, if growth influence the Turing-like system, and produced hidden PGCTPs, these might not be kept until the end of the process. We propose here three types of mechanisms aborting PGTCPs (Fig 5).

First, is the spatial decoupling between the structure initially producing the structure (e.g., the melanocytes in the epidermis or dermis of bird or mammal skin) and the structure bearing the colors (e.g., feather or hair). A completely disordered fur or down will erase the underlying

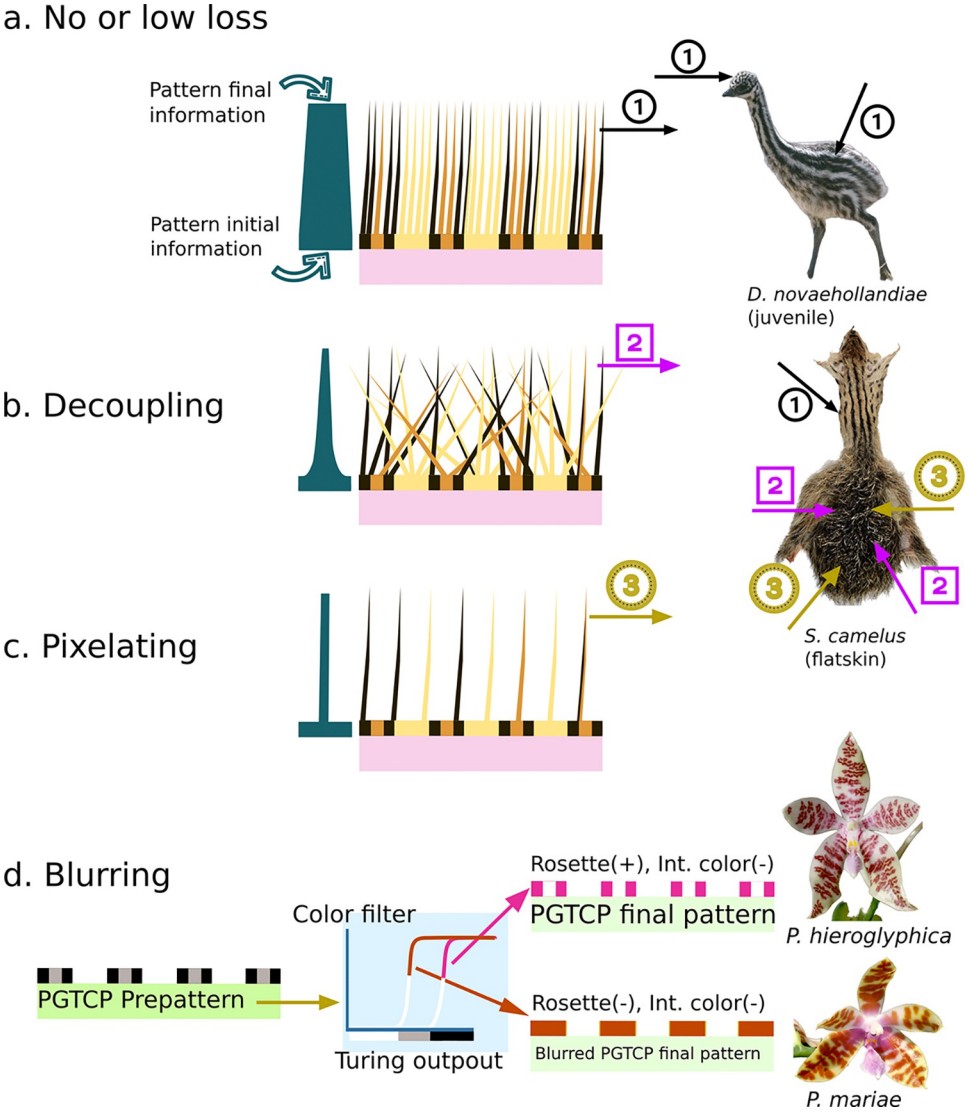

**Fig 5. Three ways to lose PGTCPs information from prepattern to final phenotype.** A) No loss. B) Decoupling C) Pixelating D) Blurring. Pictures sources: GusSar, 2013 https://commons.wikimedia.org/wiki/File:Emu_(Dromaius_novaehollandiae).jpg, Orchi, Unknown date https://fr.m.wikipedia.org/wiki/Fichier:Stanhopea_haseloviana_Orchi_001.jpg, Orchi, Unknown date https://commons.wikimedia.org/wiki/File:Phalaenopsis_mariae_Orchi_1883.jpg.

pattern, and so potential Turing-like patterns and PGTCPs. Juvenile ostriches harbor a rather disheveled down on their dorsum, compared to their head and neck. As a result, while the plucked skin shows a repeated pattern (consisting of stripes and dots), it is only observable on the head and neck (Fig 5B). But this is not the only mechanism that appears to be at work in the juvenile ostrich back, which also appears to involve what we call "pixelating".

This second mechanism is when the final-colored structure is discrete (like a feather) and characterized by a lower "resolution" than the structures producing the initial pattern (Fig 5C). It acts as a filter by mixing several contiguous system values, therefore total information is reduced, and possibly most of the pattern characteristics are lose. In fact, Juvenile ostriches present nearly submillimeter initial skin pattern resolution, but nevertheless it possesses very low density of feathers on the back, which are very large compared to the down of the neck and head. In other words, they have a very low resolution compared to the putative people underlying melanocyte skin pattern. This could lead to the loss of the geometric signal at the plumage scale.

The third type of mechanism leading to pattern loss is called blurring (Fig 5D), and we assume that it could be quite frequent in Angiosperms PGTCPs. In this case, PGTCPs could be set up, but several mechanisms like a high sensitivity to Turing morphogens leading to a high level of color production with a maximum plateau could erase the difference between "medium" zones (typically the center of a rosette) and "high" zones (the crown a rosette), resulting in a more classical spot pattern. On-Off Filters with sigmoidal kinetics like allosteric enzymes could also make invisible the effects of growth and PGTCPs.

## Experimental biases that need to be addressed to obtain a more accurate picture of PGTCPs of Eukaryotes

By definition, PGTCPs are produced by growth and individuals undergo the most growing phases during embryonic and post-embryonic development. However, most of the biodiversity textbooks and atlas focus on adult external morphology. This lack of embryonic and juvenile data clearly causes an underestimation of PGTCPs occurrences. In this study we also have noticed several times that when male/female dimorphism existed, only male individuals were represented in some textbooks. This was typically the case of *Cervus elaphus* where spotted Turing pattern of some females and juveniles was nearly missed if there had been no subsequent check.

Last but not least, inter-individual color pattern variability is very high in many species, like the Lamiaceae *Scutellaria rubropunctata*, which exhibit on its petal a range of color patterns from uniform blue or white to classic putative Turing spot patterns and striking PGTCPs like classic or open rosettes depending on the strain [27]. A most comprehensive view of PGTCPs is therefore necessarily through the examination of many individuals.

## Material and methods

### PGTCPs survey among Eukaryotes

Putative Growth Turing-like Color Patterns (PGTCPs) were classified into 4 categories, which could be combined within the same species: rosettes (and mazes of mazes), intermediate stripes, mixed colors, and line-and-dot alternation.

We checked both for classic Turing Patterns and for PGTCPs, and in these two categories four types of labels have been given be for each group: Y (Yes, if at least one species with a clear pattern has been found), M (Maybe Yes, if at least one species with a potential but not clear pattern has been found), N? (Maybe No, if no species with pattern has been found, but if

the search was non exhaustive) and N (Probably No, if no species with pattern has been found, if the search was almost exhaustive).

We chose to impose a minimum number of motifs to consider it as a PGTCP for the following reason: when the number of motifs is low, there is a high probability that other types of mechanisms than Turing-like could produce them. We decided to choose 3 motifs as a compromise, but we do not exclude the possibility that PGTCPs with only one or two motifs exist in nature, and alternatively that some of the three- or four-motifs patterns we found might not be produced by a Turing-like system. For the same reason, we chose to exclude repeated patterns based on a pre-existed repeated geometry. For example, if every petal of a flower harbors a single dot and therefore an overall periodic color pattern, a Turing mechanism might not be needed to produce it.

The complete results are available in a database accessible at the following URL (www.eukolorpatterns.cnrs.fr open to contributions and at https://www.ebi.ac.uk/biostudies/studies/S-BSST1375, DOI 10.6019/S-BSST1375). They comprised the complete list of taxa explored, the names of the candidate species and a link to a representative picture of patterns. For the almost exhaustive surveys of mammals and orchids, lists of species were respectively extracted from ASM Mammal Diversity database (https://www.mammaldiversity.org/, January 2022), and COL, Catalogue of Life (https://www.catalogueoflife.org/).

We use multiple sources for photographic or textual proofs attesting a classic Turing pattern or a PGTCP, Handbook of the Mammals of the World (HMW), Volume 1 to 9, Lynx Edicions for mammals and http://www.orchidspecies.com/ for orchids, with additional sources from the scientific literature when a photograph was not available. When possible, in mammals, juvenile patterns were also looked at, although the vast majority of sources are limited to the adult, and most often the male. In some species, sexual dimorphism resulted in the presence of a Turing pattern only in the female (e.g., *Cervus elaphus*), and in others, only the juveniles showed a PGTCP (e.g., the wild boar *Sus scrofa*). Many examples of PGTCPs may have been missed due to poor sources on juvenile phenotypes.

The lists of orders for animals and families for Angiosperms (or higher or lower taxonomic level if order/family did not exist) were extracted from COL, Catalogue of Life (https://www.catalogueoflife.org/). On a case-by-case basis but not exhaustively, corrections were made if scientific literature no longer recognized the taxon. Again, the sources of photography were multiple, scientific literature was preferred, then online encyclopedia Wikipedia (https://en.wikipedia.org/) and Wikimedia Commons (https://commons.wikimedia.org), eventually alternative sources (through search engines) with keywords such as "pattern", "spotted", "stripes", when the first three were not conclusive.

## PGTCPs mapping on phylogenetic trees

Phylogenetic relationships between species were built with phyloT (V2, https://phylot.biobyte.de/) based on NCBI taxonomy. Unrecognized taxa were resolved by hand. Phylogenetic trees were plotted and annotated in iTOL (Interactive Tree Of Life, version 6.5.8, https://itol.embl.de/). Tree leaves were name after the found species name if at least one example was found, or by the name of the order/family if not.

## PGTCPs modelling

A custom Turing model was built in Processing 4.0 software language (https://processing.org/), based on De Gomensoro Malheiros et al. equations [20].

**An example of code can be found at:** (https://github.com/PierreGalipot/PGTCPs/blob/main/Leopard.pde)

Master equations:

$$a(t+1) = \frac{a(t)}{1+g*dt} + \left(16 - \frac{a(t)}{1+g*dt} * \frac{b(t)}{1+g*dt} + \frac{d(t)}{(1+g*dt)*(1+g*dt)} *r*s*\nabla 2a\right) *dt$$

$$b(t+1) = \frac{b(t)}{1+g*dt} + \left(\frac{a(t)}{1+g*dt} * \frac{b(t)}{1+g*dt} - \frac{b(t)}{1+g*dt} - 12 + \frac{d(t)}{(1+g*dt)*(1+g*dt)} *s*\nabla 2b\right) *dt$$

Where a and b are the morphogens concentrations, g is the growth rate (of the surface), r and s are ratio and scale parameters of De Gomensoro Malheiros et al. equations [20], $\nabla 2a$ and $\nabla 2b$ the Laplacian operators. The discrete Laplacian operator used in the simulations is the standard nine-point stencil, covering a Moore neighborhood of unitary size: [[1, 4, 1], [4, −20, 4], [1, 4, 1]] /6.

**The code of master equations:**

float newvaluea = oldvaluea/(1+g*dt) + (16—oldvaluea*oldvalueb/((1+g*dt)*(1+g*dt)) + d*r*s*laplaceA)*dt;

float newvalueb = oldvalueb/(1+g*dt) + (oldvaluea*oldvalueb/((1+g*dt)*(1+g*dt))—oldvalueb/(1+g*dt) - 12 + d*s*laplaceB)*dt;

where g is the growth rate. Growth was indirectly implemented by its effect on diffusion (via s parameter) and its effect on morphogens concentration (via dilution, on parameters a and b).

g is equal to zero until a certain point (t = 10000 in all of our simulations), meaning that a classic Turing system starts to organize before growth starts.

In S1C Fig, morphogens dilution due to growth was ignored in these simulations, due to the low growth rate compared to system dynamics.

**Fixed parameters for all simulations presented in this study:**

dt = 0.002

minA = 0

maxA = 4.4

minB = 3.5

maxB = 100

To avoid boundary effects, left and right border were connected, as well as up and down borders.

Variable parameters for simulations presented in this study are summarized in Table 2:

**Table 2. Simulations parameters.**

| Simulation | Leopard | Emu | Pomacanthus | Ictidomys | Tetraodon |
|---|---|---|---|---|---|
| Initial conditions | Random noise | One middle stripe | One middle stripe | Periodic stripes | Random noise |
| s | 6 | 10 | 10 | 6 | 10 |
| r | 30 | 6 | 6 | 30 | 8 |
| g | 0.05 | 0.05 | 0.05 | 0.01 | 0.075 |
| dt | 0.002 | 0.002 | 0.002 | 0.002 | 0.002 |
| Timing of growth start | 10000 | 10000 | 20000 | 1000 | 10000 |

## PGTCPs occurrences morphometrics

$$growth\_factor(pixel) = \frac{\sum_{i=1}^{22} \frac{growth\_factor(segment(i))}{distance(pixel,segment(i))^2}}{\sum_{i=1}^{22} \frac{1}{distance(pixel,segment)^2}}$$

For Burchell's zebra (*Equus quagga burchellii*), we hypothesized that its external embryonic development was close to that of the horse (*Equus ferus caballus*, whose embryogenesis is well documented. We selected a panel of landmarks spaced throughout the body delineating 22 segments whose lengths were measured at each developmental stage. For each segment, the relative increase in size was calculated (growth factor), allowing us to associate with each point of the tissue an estimated value of local growth depending on its distance to each of the 22 segments and on each segment growth factor.

A color code was then attributed for each pixel and plotted in Processing 4.0 software language (https://processing.org/).

Code can be found at this URL: https://github.com/PierreGalipot/PGTCPs/blob/main/Occurences_PGTCPs_VS_Growthmaps.pde

This growth map was then compared to a zebra map of PGTCPs occurrence build by hand by reporting tissue zones where PGTCPs are observed.

In parallel, for all Feliformia species which presented PGTCPs, we checked their location on the body surface, and compared back, head and legs patterns for at least three photographs per species. Additionally, six species representative of other Mammalian lineages and PGTCPs diversity were checked (*Pusa hispida* rosettes, *Tapirus indicus* (juvenile) and *Ictidomys tridecemlineatus* line-and-dots alternations, *Lemniscomys barbarus* intermediary stripes, *and Giraffa camelopardalis* mazes of mazes) together with the only avian species presenting clear PGTCP (the emu *Dromaius novaehollandiae* rosettes). For all these species, when PGTCPs where restricted to an area, they were located on the back, which is the body region knowing the strongest growth during Mammalian development.

## Reverse method morphometrics

Two data were needed to estimate the timing of the pattern implementation: i) tissue morphometrics during development, for this purpose distances between landmarks were used as a proxy of the shape and ii) the theoretical shape of the tissue at the formation of the pattern, obtained by using adult pattern and applying corrections from the number of patterns present on each distance. For example, if the neck-tail distance of an adult leopard is x times longer than the head, but there are y times fewer motifs along it, this means that at the time of pattern formation the relative size of the neck-tail distance was 1/(x*y) that of the head. By repeating this for independent distances on the tissue, if the estimations tend to converge to a specific developmental stage, we could make the reasonable assumption that this is the timing of the end of Turing pattern production.

We used the cat (*Felis silvestris*) and leopard models, the first having a well described embryogenesis for which morphometric measurements are applicable, the second for its non-canonical patterns, while being morphologically close to the first at least until its juvenile stages. We selected 11 distances representative of external morphology (head, trunk, tail, neck, limbs), and measurable from early embryonic stages to adults. All these distances were plotted relative to head length (S1B Fig) during development. The head was chosen because its dynamics is far from most other distances, which allows for maximum contrast in relative distances.

Motifs counts of the adult cat and adult leopard were used to calculate theoretical values for each distance, and by graphical reading, the most likely approximate theoretical stage to which the distance refers was estimated and represented by ellipses, whose size was representative of the relative distance range covered during development (in this case, tail and trunk have the most contrasted dynamics compared to the head). This technique was also applied to PGTCPs, considering either that a rosette (or equivalent) was a single motif (to calculate the timing of production of the first generation of dots), or by considering its constituent sub-patterns (to date the second generation of motifs).

From each timing prediction, the wavelength of the pattern at its formation can be calculated, since the number of patterns along its distance is known and the distances are also known in an absolute way by morphometry. In the same way, if these predictions converge towards a value, we can assume that this value is close to the real value of the wavelength of the initial pattern.

For leopard dots, predicted timings are rather scattered, but all estimations point to late timings, post stage 22. This dispersion may have one or several non-mutually exclusive causes such as: approximation of landmark distances and positions, approximation in the motif counting, differences in external morphology between cat and leopard, non-trivial relationship between number of motifs and tissue size or finally, different timing of pattern formation depending on the tissue location. If we focus on the two distances that undergo the greatest increase in relative size with respect to the head (trunk and tail), which *a priori* are the two most reliable values for estimation, they suggest stage 22 as the stage of ending of phase 1 of patterning (formation of the 1st generation of dots, which will give the rosettes).

For all simulations, whatever the growth rate, we observe a discrepancy between the number of patterns expected and the number of patterns observed, which can be initially attributed to the inertia of the system (the time taken to respond to the perturbation). Nevertheless, if we let the system run long enough after the end of growth, we observe very clearly that the final number of patterns is lower than the expected one, by about 15% for all growth rates in our example. This means that when the Turing system undergoes a first pattern generation, then responds to growth, it imposes constraints which prevent the system from reaching its maximum number of motifs at the equilibrium.

## Reverse method modelling

To test the prediction validity, and how well patterns can be used as predictors of morphometry, we performed simulations of PGTCPs formation and counted the number of motifs obtained over time and compared to the theoretical ones. To calculate the theoretical value of the motifs number, simulations were run without growth and at different tissue sizes. As expected, the final number of motifs is proportional to the tissue size, for a given set of Turing system parameters. Different growth rates were simulated (0.01, 0.02, 0.05, 0.1, 0.2) starting from a tissue size = 1 with different maximum sizes (1/0.4, 1/0.25, 1/0.185). The number of iterations was between t = 75000 and t = 100000 and a screenshot was taken every t = 250. The number of motifs was counted in an automated way with a custom macro on Fiji software (version 2.3.0) based on the "Find maxima" process, after having verified the validity of the pattern detection algorithm in several representative situations.

Although we let the simulations continuing after growth had stopped, in order to observe the behavior of the system, the most important timing is the moment when the tissue reached its final size. The difference between the number of motifs observed at this timing and the expected number of motifs was calculated and compared between different values of growth rate and final size.

Several things could counterbalance this discrepancy. First, it should be noted that in cases where the growth rate is too high, the system does not have time to produce growth patterns. This means that if growth patterns are observed, the growth rate is necessarily reduced, and therefore the shift with the expected number of patterns should be lower.

In our models, the "delta" was between 16 and 20%, for growth rates low enough for growth patterns production. In order to determine if these values were verified for other conditions, we carried out other simulations with higher final size values. We then obtained 'delta' values all between 10 and 20% (not shown). Without knowing what the relationship is between tissue growth rate and response rate of biological systems, we therefore propose to use a correction of +15%, +/-5% to the number of motifs counted in presence of rosettes, which should bring the estimated value closer to the theoretical value and improve the accuracy of our reverse method.

## Artificial landmarks and in vivo monitoring and estimation of tissue local growth

We adapted a photogrammetry method [49] to an *in vivo* flower bud of *Phalaenopsis hieroglyphica*. Points were disposed by hand with a laboratory waterproof ink marker pen. Photographs have been taken with an Olympus OMD E-M10 14-42mm camera. For preliminary test of geometric morphometrics, landmark coordinates were manually recorded using tpsDig2 [50] and processed following standard procedures with tpsRelw [50]. Landmark configurations were optimally aligned by generalized Procrustes analysis [51], which extract shape data by filtering out differences in scale, location, and orientation among raw landmark configurations. The patterns of shape variation were explored using the principal component analysis (PCA) [52].

## Fourier analyses of PGTCPs

Image Fourier analyses were performed on pictures from i) filtered classic Turing and PGTCPs simulations and ii) filtered photographs of real species. Morphogens dilution due to growth was ignored in these simulations, due to the low growth rate compared to system dynamics. To reduce noise and artifacts due to pixelating, we transform colored pictures using GIMP software (version 2.10.30) with several color filters (Image>Mode>Indexed>Black and White 1-bit palette for simulations, and Colors>Hue-saturation/Brightness-Contrast/Levels for photographs), to obtain pure black and white (no grey) images of size 500*500 pixels. We built an algorithm to calculate the Discrete Fourier Transform (DFT) to visualize the frequency domain in a 2D graph, at the same size of the original image, with a pixel intensity greyscale. Each particular frequency is quantified by a complex number, and we represented their magnitude (square root of the sum of the square of the real and square of imaginary parts), which is the data that contains the most information. Classically, (0,0) frequency, i.e., the image mean was centered and the more the point is far for the center, the more the frequency is high (the most the wavelength is short). The Fourier analyses permits to visualize and quantify the pattern periodicities to pinpoint the presence of multiple frequencies and fractalness of some patterns, and to precisely compare patterns between them. For a more appropriate visualization, we apply a log10 transformation to the magnitude values, then divided them by the max value among the frequency magnitudes (or alternatively for the maximum value of a given series of pictures in order to normalize), and a threshold corresponding to 1/20 or 1/100 the maximum magnitude value to eliminate the less important frequencies.

Three screenshots from three steps of a PGTCP rosette simulation and four pictures from three species were used in this analysis: a picture of an *in vivo* blue-ringed octopus

*Hapalochlaena lunata* for its ring PGTCP (a particular kind of rosette pattern), a picture of a flatskin of the leopard *Panthera pardus* for its rosettes, and in-vivo pictures of adult and juvenile of *Tetraodon mbu* for its maze of mazes of mazes (a PGTCP with first, second and third order motifs). In real species, to study the periodicity characteristics of PGTCPs compared to classic ones, we artificially erased the effects of growth by -as an example- filling the holes to transform rings and rosettes into dots, from which they putatively derive. For *Tetraodon mbu* adult pattern, we generate first a $2^{nd}$ order pattern by erasing the littlest motifs, and then a $1^{st}$ order pattern by erasing the intermediary motifs. The latter pattern is similar to the juvenile one in terms of motifs geometry and density, suggesting the continuity between the two. For all Fourier analysis, the resulting DFT graphs show circular symmetry of the spectra, marking the absence of preferential directionality of the studied patterns (contrary for example to a parallel striped pattern). Rosette simulation images DFT show an evolution of the frequency distribution during growth, which is marked by the widening of the spectrum (small frequencies are more represented after growth, as the simulation is set to keep the same image size). The intermediary picture (t = 23500) has the combined spectrum characteristics of the early (t = 20000, pre-growth) and the late (t = 30000) patterns. Despite limitations characterizing photographs of real patterns (like the low number of motifs for *Hapalochlaena lunata* or the pattern deformations due to skin folding in 3 dimensions and photograph perspectives), real species patterns show a similar trend, *i.e.*, a more frequency-rich pattern for PGTCPs due to the additivity of the periodicity of the initial motifs and the ones that arise during growth.

Code is available at this URL: https://github.com/PierreGalipot/PGTCPs/blob/main/Fourier_transform.pde

## Code availability

All codes used for this study (Figs 2–4, S1 and S2 Figs) are available at https://github.com/PierreGalipot/PGTCPs

## Supporting information

**S1 Fig. PGTCPs can be a source of information thanks to a reverse method.** (**A**) Left: examples of distortions in classic Turing pattern (*Mimulus sp.*) and a PGTCP (*Panthera pardus*) Right: reverse method principles and assumptions. (**B**) Dynamics of relative size of various distances to head size during development of the cat (*Felis silvestris* stages 14 to 23) and leopard (*Panthera pardus*, stage 24 and 25) and predicted classic and PGTCPs pattern formation. (**C**) Simulation of dynamics of motifs number during pattern formation under tissue growth. Y-axis: number of motifs simulated at different growth rates (solid lines), compared to the theoretical number of motifs (dashed lines). Up left and down right: 4 details of pattern phenotype through growth for g = 0.05, at different times (1: early, 2: pattern at the start of growth, 3: pattern just before growth stops, 4: late pattern at the equilibrium). (**D**) Tissue growth monitoring with hand-written landmarks on the tissue, example of *Phalaenopsis hieroglyphica* sepal outer epidermis. Upper panel: Sequence of landmarked flower bud. Lower panel: Left, four views extracted from the photogrammetry model of *P. hieroglyphica* flower bud (at Day 6). Right, preliminary 2D geometric morphometrics landmarks assessment and comparison between young stage (Day 6, $1^{st}$ and $3^{rd}$ pictures) and final stage (Day 41, $2^{nd}$ picture). Fourth picture: grid deformation between young and late stage after Procrustes analysis. Pictures sources: Patrick Giraud 2006 https://commons.wikimedia.org/wiki/File:Namibie_Etosha_Leopard_01edit.jpg, Hugo.arg 2007 https://commons.wikimedia.org/wiki/File:Mimulus001.JPG.
(TIF)

**S2 Fig. Discrete Fourier Transforms (DFT). (A**) Filters applied to PGTCPs simulations output to obtain a 1-bit image and the respective frequency magnitude spectra. (**B**) DFT of blue-ringed octopus *Hapalochlaena lunata* PGTCPs (first two panels) and their modified version without PGTCP (last two panels). (**C**) DFT of leopard (*Panthera pardus*) flastskin PGTCPs (first two panels) and their modified versions without PGTCP (last two panels). (**D**) DFT of simulated PGTCPs (first three panels, log magnitude, no threshold) and *Tetraodon mbu* PGTCPs (last four panels) and their modified version without PGTCP (last two panels, log magnitude, no threshold). Pictures sources: Rickard Zerpe, 2019 https://commons.wikimedia. org/wiki/File:Greater_blue-ringed_octopus_(Hapalochlaena_lunulata)_(48272090161).jpg, SagarPaudel68 2016 https://commons.wikimedia.org/wiki/File:Leopard_skin.jpg.
(TIF)

**S3 Fig. Classic Turing and PGTCP survey methods.** (A) Screenshot of PGTCP database. (B) Classic Turing and PGTCP survey flowchart. (C) Criteria and examples of candidates. A minimal number of three motifs is required to consider it as a good candidate for a PGTCP, as one or two motifs patterns might be produced by other mechanisms than the Turing system. Pictures sources: Derek Keats 2011 https://commons.wikimedia.org/wiki/File:Stapelia_gigantea_ (5538239090).jpg, Greg McFal 2010, https://fr.wikipedia.org/wiki/Fichier:Regal_Sea_ Goddess_Nudibranch.jpg.
(TIF)

# Acknowledgments

I deeply thank Dr. Marianne Elias and Dr. Sylvain Gerber for their corrections, remarks and advice about all phases of the project.

I deeply thank Prof. Marcelo de Gomensoro Malheiros from Centro de Ciências Computacionais, Brasil for his valuable remarks and suggestions concerning the numerical simulations.

I deeply thank Dr. Bart Buyck (ISYEB, MNHN, France), Dr. Tsuyoshi Hosoya, Dr. Jean-Philippe Rioult (UniCAEN, France) and Pr. Marc-Andre Selosse (ISYEB, MNHN, France) for the valuable mycological discussions concerning the links between Fungi, colors and Turing-like patterns.

I deeply thank Dr. Violaine Llaurens (ISYEB, MNHN, France) for her precious advice on the organization of the project and her remarks on the notions of Evolution mentioned in the article.

I deeply thank Dr. Visotheary Ung (ISYEB, MNHN, France) for having helped to set up the website hosting the database and for her precious advice on the data formatting.

I deeply thank Marion Leménager and Hirokazu Tsukaya for their help concerning the photogrammetry experiments on *Phalaenopsis hieroglyphica*.

I deeply thank Émilie Dupré for her work on classic Turing and PGTCPs pattern recognition by deep learning, which will help complete the database and make it as exhaustive as possible in the future.

I deeply thank Florian Jabbour and Catherine Damerval for their advice and corrections on the ultimate version of the project.

# Author Contributions

**Conceptualization:** Pierre Galipot.

**Data curation:** Pierre Galipot.

**Formal analysis:** Pierre Galipot.

**Investigation:** Pierre Galipot.

**Methodology:** Pierre Galipot.

**Visualization:** Pierre Galipot.

**Writing – original draft:** Pierre Galipot.

**Writing – review & editing:** Pierre Galipot.

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
