## [Decision Letter · Decision Letter 0]

23 Jan 2024

PONE-D-23-43374And Growth on Form? How tissue expansion generates novel shapes, colours and enhance biological functions of Turing colour patterns of EukaryotesPLOS ONE

Dear Dr. Galipot,

Thank you for submitting your manuscript to PLOS ONE. After careful consideration, we feel that it has merit but does not fully meet PLOS ONE’s publication criteria as it currently stands. Therefore, we invite you to submit a revised version of the manuscript that addresses the points raised during the review process.

We look forward to receiving your revised manuscript.

Kind regards,

Hualin Fu

Academic Editor

PLOS ONE

Journal Requirements:

Reviewers' comments:

Reviewer's Responses to Questions

**Comments to the Author**

1. Is the manuscript technically sound, and do the data support the conclusions?

Reviewer #1: Yes

Reviewer #2: Yes

Reviewer #3: Partly

2. Has the statistical analysis been performed appropriately and rigorously? 

Reviewer #1: N/A

Reviewer #2: I Don't Know

Reviewer #3: N/A

3. Have the authors made all data underlying the findings in their manuscript fully available?

Reviewer #1: Yes

Reviewer #2: Yes

Reviewer #3: Yes

4. Is the manuscript presented in an intelligible fashion and written in standard English?

Reviewer #1: Yes

Reviewer #2: Yes

Reviewer #3: Yes

5. Review Comments to the Author

Reviewer #1: This paper is well-written from a language point of view, but I find it lacking in its rigour and concreteness. My biggest point of this is that I’m still not really sure what the difference between a classical and a PGTCP is. The best definition I can see appears on page 11, which is too far into the paper. So I spend a long time not really understanding what the paper is talking about.

Saying that, my understanding is that a classical Turing system is a model of interacting agents where the interactions, movement mechanisms and domain features are all static, whereas a PGTCP is anything else.

From this point I’m not really sure as to the point of the paper.

There are papers out there which already highlight the diversity of Turing patterns

@Article{Krause-2020-FOP,

author = {Krause, A. L. and Klika, V. and Woolley, T. E. and Gaffney, E. A.},

title = {From one pattern into another: analysis of Turing patterns in heterogeneous domains via WKBJ},

number = {162},

pages = {20190621},

volume = {17},

journal = {J. Roy. Soc. Interface},

publisher = {The Royal Society},

year = {2020},

}

@Article{Krause-2018-HIS,

author = {Krause, A. L. and Klika, V. and Woolley, T. E. and Gaffney, E. A.},

title = {Heterogeneity induces spatiotemporal oscillations in reaction-diffusion systems},

number = {5},

volume = {97},

journal = {Phys. Rev. E},

year = {2018},

}

Moreover, the paper ignores a whole section of mechanisms through the application of noise to the systems, which also include growth.

@Article{Woolley-2011-PSS,

Title = {Power spectra methods for a stochastic description of diffusion on deterministically growing domains},

Author = {Woolley, T. E. and Baker, R. E. and Gaffney, E. A. and Maini, P. K.},

Journal = {Phys. Rev. E},

Year = {2011},

Month = {Aug},

Number = {2},

Pages = {021915},

Volume = {84},

DOI = {10.1103/PhysRevE.84.021915},

Numpages = {15},

Publisher = {American Physical Society}

}

@Article{Woolley-2011-PSS2,

Title = {Influence of stochastic domain growth on pattern nucleation for diffusive systems with internal noise},

Author = {Woolley, T. E. and Baker, R. E. and Gaffney, E. A. and Maini, P. K.},

Journal = {Phys. Rev. E},

Year = {2011},

Month = {Oct},

Number = {4},

Pages = {041905},

Volume = {84},

DOI = {10.1103/PhysRevE.84.041905},

Issue = {4},

Numpages = {13},

Publisher = {American Physical Society},

URL = {http://link.aps.org/doi/10.1103/PhysRevE.84.041905}

}

@Article{Woolley-2012-EIS,

Title = {Effects of intrinsic stochasticity on delayed reaction-diffusion patterning systems},

Author = {Woolley, T. E. and Baker, R. E. and Gaffney, E. A. and Maini, P. K. and Seirin-Lee, S.},

Journal = {Phys. Rev. E},

Year = {2012},

Number = {5},

Pages = {051914},

Volume = {85},

Publisher = {APS},

Timestamp = {2013.06.28}

}

Finally, there's even work on how to design models to ensure that you get patterns that you want

@Article{Woolley-2021-BTS,

author = {Woolley, T. E. and Krause, A. L. and Gaffney, E. A.},

title = {{Bespoke Turing Systems}},

number = {5},

pages = {1--32},

volume = {83},

journal = {Bulletin of Mathematical Biology},

publisher = {Springer},

year = {2021},

}

All in all, this is a nice biological review on the diversity of applications of extending Turing systems. This is needed for the biological community as well as for the mathematical community, but I don’t really see a conclusion beyond Turing systems can account for patterns beyond stripes, spots and labyrinthine if the mechanism is complicated in some manner… which we already knew. Thus, I suggest the author tries to find a stronger conclusion, or some different insight to finish on.

Reviewer #2: The concept of reaction-diffusion mechanisms as a means to explain periodic patterns in nature has been prevalent for over seven decades. This elegant mathematical theory proposed by Alan Turing stands as a cornerstone in our understanding of natural patterns. The authors of this study have honed in on a particularly pervasive yet specific phenomenon: the coupling of reaction-diffusion patterning with tissue growth, termed here as 'putative growth Turing-like color patterns' (PGTCPs).

This paper is commendable for its unique perspective on how tissue expansion influences reaction-diffusion patterning. It delves into several intriguing facets of this subject. The authors commence their exploration with a comprehensive, large-scale screening to determine the prevalence of PGTCPs across various species. This is followed by an insightful elucidation of the formation mechanisms underlying four distinct sub-classes of PGTCPs. The comparative analysis of PGTCP manifestations in different parts of the same organism, in relation to the varied tissue growth dynamics at these sites, is particularly noteworthy. The hypothesis presented towards the end, suggesting the evolutionary role of PGTCPs in signaling and camouflage, adds an intriguing dimension to the study.

However, as a manuscript intended for a general readership, it requires further refinement. The figures and explanatory sections particularly need enhancement for better clarity. Providing clear, concise key information and essential details is crucial to aid readers in grasping the narrative flow and the methodologies employed in the experiments. Such improvements would significantly augment the paper's accessibility and comprehension.

Major revisions:

A fundamental question addressed in this study is the dependency of the final pattern outcome on the rate of tissue growth relative to the dynamics of pattern formation mediated by the Turing mechanism. The authors propose a hypothesis regarding the formation of four typical putative growth Turing-like color patterns (PGTCPs), suggesting that tissue expansion plays a decisive role only during a specific phase of patterning. According to this hypothesis, there are three distinct phases: Phase I: The tissue remains unchanged, allowing the initial pattern to form and stabilize. Phase II: Tissue expansion becomes significant, catalyzing further evolution of the pattern and the emergence of PGTCPs. Phase III: Tissue expansion slows down or ceases entirely, preserving the newly emerged PGTCPs without further alteration. This critical explanation is currently located in the legend of Supplementary Figure 1C. However, given its importance in understanding the interplay between tissue growth and patterning, it would be more appropriate to incorporate this discussion into the main text, preferably in the early sections of the manuscript.

In the 'Materials and Methods' section, particularly in the PGTCPs modeling session, it is imperative for the authors to provide essential information about the models. This should include the modeling assumptions, master equations for each circuit species, simulation methods, and expected outputs. Such details are crucial for comprehensively understanding and replicating the study's findings.

Minor revisions:

In the figure legends, the authors should provide more comprehensive details about the key steps of each simulation experiment to facilitate a better understanding of the simulation results among readers.

Figure 1A:

Do these four categories encompass the entire spectrum of PGTCPs?

Figure 1C & 1D:

What criteria are used to differentiate classic Turing patterns from PGTCPs? Is this classification based on visual assessment of stable patterns in standard adult species? For instance, are spots, lines, and mazes classified as classic Turing patterns, while rosettes, line-and-dot alternations, mixed colors, and intermediate bands are categorized as PGTCPs?

Could you clarify the distinction between the categories “Yes” and “Maybe”?

Figure 2A:

Can you describe how the tissue expands over time? Does it begin expanding constitutively from t = 0?

Intuitively, a very slow expansion rate might lead to a regular spot pattern. If the expansion rate is significantly higher, how does the rosette pattern evolve dynamically? Do "disrupted rosettes" emerge as a pattern?

Figure 2C:

What is the time sequence for the six panels shown?

How does tissue expansion correlate with pattern formation dynamics?

Figure 2D:

What does the dotted line plot represent? Below the lower t = 0 panel, two color sets are used for the dots. Please specify their meanings.

The dotted line plot below the second row is too dim to discern.

In the lower t = 0 panel, are the stripes pre-assigned as initial conditions for the simulation?

Is it correct to assume that in these simulations, tissue expands only along the x-axis?

Figure 2F:

This figure appears to have the same issues as mentioned for Figure 2C.

Figure 3B:

Please specify what the star marks and lines indicate.

Supplementary Figure 1C:

Label the y-axis to provide clarity

Reviewer #3: Please see attached comments.

6. PLOS authors have the option to publish the peer review history of their article (what does this mean?). If published, this will include your full peer review and any attached files.

Reviewer #1: No

Reviewer #2: No

Reviewer #3: **Yes: **Andrew Krause

---

## [Author Response · Author response to Decision Letter 0]

3 Apr 2024

To : the Editors of PLOS One At Rennes, March, 28th 2024

Subject : Rebutal letter

Dear Editors,

I would like to thank the reviewers for their generous comments on the manuscript and I have edited the manuscript to address their concerns. I believe and hope that the manuscript is now suitable for publication in PLOS One. 

Please find below my answers to every point mentioned by the reviewers. In the rebuttal letter file, their reviews are in bold and the answers in yellow-brown. Quotes of corrected texts are in yellow-brown and in italics.

I remain at your disposal for any information, 

Very sincerely, 

Dr. Pierre Galipot

ECOBIO

Université de Rennes - Campus Beaulieu

Rennes, FRANCE

Reviewer #1

“My biggest point of this is that I’m still not really sure what the difference between a classical and a PGTCP is. The best definition I can see appears on page 11, which is too far into the paper. So I spend a long time not really understanding what the paper is talking about.

Saying that, my understanding is that a classical Turing system is a model of interacting agents where the interactions, movement mechanisms and domain features are all static, whereas a PGTCP is anything else.

From this point I’m not really sure as to the point of the paper.”

Thanks for this useful comment. 

I added a complete definition of PGTCPs in the introduction section.

-> lines 87 to 91 “Here, we consider a particular color pattern as a PGTCP when it possess all of these characteristics: i) periodic, at least at the local scale, ii) non-based on a pre-existing periodicity (e.g. body segments or petals), iii) with at least three motifs (e.g. three stripes or rosettes) and iv) with at least one motif not belonging to the classical range of Turing geometries (dots, stripes, mazes), i.e. line-and-band alternations, rosettes for example.”

“There are papers out there which already highlight the diversity of Turing patterns

@Article{Krause-2020-FOP,

[...]

@Article{Krause-2018-HIS,[...]”

Thank you for these references.

I will not comment on all parts of these articles, but only on what concerns PGTCPs. The first article (Krause 2020) proposes a way to produce rosette patterns of jaguar without necessarily growth (Figure 1), from a complex pre-pattern. I agree that some examples of PGTCPs might not be produced by growth in nature.

Anyway, the purpose of our article was not to claim that growth was the only way to produce PGTCPs, but the most likely for the majority of examples. We produced several types of arguments summarized in Figure 3.

As far as I understand the second article, it does not completely deal with PGTCPs formation.

To emphasize on the fact that growth is not the only way to produce PGTCPs, we add explanations in the introduction parts, and references, including (Krause 2020).

-> lines 75 to 77 “Most of those models perfectly reproduced leopard rosettes by adding the most probable missing ingredient: tissue growth during pattern formation. Other types of missing ingredients have been described, like complex pre-patterns (21).”

“Moreover, the paper ignores a whole section of mechanisms through the application of noise to the systems, which also include growth.

@Article{Woolley-2011-PSS,

Title = {Power spectra methods for a stochastic description of diffusion on deterministically growing domains},

[...]”

“@Article{Woolley-2011-PSS2,

Title = {Influence of stochastic domain growth on pattern nucleation for diffusive systems with internal noise},

URL = {http://link.aps.org/doi/10.1103/PhysRevE.84.041905}

[...]

Thank you for bringing these articles to my attention.

These articles show another effect of growth in spatially heterogeneous patterns formation. Without Turing systems and only thanks to growth and stochasticity, the authors showed that heterogeneous patterns could arise. The resulting patterns are heterogeneous like Turing ones and PGTCPs but lack periodicity and some particular geometries, they are then not sufficient to explain the diversity of patterns evoked in our article.

@Article{Woolley-2012-EIS,

Title = {Effects of intrinsic stochasticity on delayed reaction-diffusion patterning systems},”

Thank you for bringing this article to my attention.

The authors showed that taking stochasticity into account could accelerate in some cases Turing pattern formation, compared to classic deterministic models. Growth is not simulated in this article so it is difficult to assess a particular role of stochasticity in PGTCPs formation.

“Finally, there's even work on how to design models to ensure that you get patterns that you want

@Article{Woolley-2021-BTS,”

This article proposes a nice and pedagogical method on how to build a pattern with particular desired motifs (spots, stripes or labyrinthine) from a desired set of ranges of Turing parameters. Nevertheless, it does not encompass PGTCPs motifs evoked in our paper (rosettes, intermediate colors, etc.) even if the juvenile of Tapir, presenting alternation of stripes and dots is shown in Figure 7. More precisely, simulation of coexistence of stripes and spots are shown (Figure 6), but not alternation of stripes and spots. In our paper, we propose that growth is not needed to explain simple coexistence but on the contrary is needed to produce an alternation. Therefore, this article does not sufficiently cover PGTCPs formation.

“All in all, this is a nice biological review on the diversity of applications of extending Turing systems. This is needed for the biological community as well as for the mathematical community, but I don’t really see a conclusion beyond Turing systems can account for patterns beyond stripes, spots and labyrinthine if the mechanism is complicated in some manner… which we already knew. Thus, I suggest the author tries to find a stronger conclusion, or some different insight to finish on.”

Thanks for this important remark. To better explain the purpose of the article and its conclusions, I rephrase some parts to be clearer:

-> lines 103 to 110: “In summary, we have shown that the diversity of PGTCPs extends far beyond leopards and a few other animals, to thousands of species, including angiosperms. In addition, while some of the effects of growth were already known (like rosette formation), we used simulations to show other possible effects that had not yet been described and were biologically plausible. We then produced a large number of concordant clues of different kinds, showing the very high probability of growth as the cause of PGTCPs. As the great diversity of PGCTPs may suggest potential biological functions, we have described and discussed several of them, with supporting arguments”

—---------------------------------------------------

Reviewer #2: The concept of reaction-diffusion mechanisms as a means to explain periodic patterns in nature has been prevalent for over seven decades. This elegant mathematical theory proposed by Alan Turing stands as a cornerstone in our understanding of natural patterns. The authors of this study have honed in on a particularly pervasive yet specific phenomenon: the coupling of reaction-diffusion patterning with tissue growth, termed here as 'putative growth Turing-like color patterns' (PGTCPs).

This paper is commendable for its unique perspective on how tissue expansion influences reaction-diffusion patterning. It delves into several intriguing facets of this subject. The authors commence their exploration with a comprehensive, large-scale screening to determine the prevalence of PGTCPs across various species. This is followed by an insightful elucidation of the formation mechanisms underlying four distinct sub-classes of PGTCPs. The comparative analysis of PGTCP manifestations in different parts of the same organism, in relation to the varied tissue growth dynamics at these sites, is particularly noteworthy. The hypothesis presented towards the end, suggesting the evolutionary role of PGTCPs in signaling and camouflage, adds an intriguing dimension to the study.

However, as a manuscript intended for a general readership, it requires further refinement. The figures and explanatory sections particularly need enhancement for better clarity. Providing clear, concise key information and essential details is crucial to aid readers in grasping the narrative flow and the methodologies employed in the experiments. Such improvements would significantly augment the paper's accessibility and comprehension.

Major revisions:

A fundamental question addressed in this study is the dependency of the final pattern outcome on the rate of tissue growth relative to the dynamics of pattern formation mediated by the Turing mechanism. The authors propose a hypothesis regarding the formation of four typical putative growth Turing-like color patterns (PGTCPs), suggesting that tissue expansion plays a decisive role only during a specific phase of patterning. According to this hypothesis, there are three distinct phases: Phase I: The tissue remains unchanged, allowing the initial pattern to form and stabilize. 

Phase II: Tissue expansion becomes significant, catalyzing further evolution of the pattern and the emergence of PGTCPs. Phase III: Tissue expansion slows down or ceases entirely, preserving the newly emerged PGTCPs without further alteration. This critical explanation is currently located in the legend of Supplementary Figure 1C. However, given its importance in understanding the interplay between tissue growth and patterning, it would be more appropriate to incorporate this discussion into the main text, preferably in the early sections of the manuscript.

Thanks for this important remark. 

As the literature is very exhaustive concerning the mathematical approach on Turing pattern formation, I decided not to focus to much on dynamics, also for two other complementary reasons: 1) we usually simulate Turing patterning with reaction-diffusion equations with two morphogens, but the real biological systems could be very different and much more complex, leading to any discussion on dynamics partly obsolete. Because I wanted to focus on biological clues in this article, I generally tried to avoid discussion about mathematical relationships. 2) The other reason is that growth could generate PGTCPs under many conditions. For example, in the study, I decided in simulations to begin with a first phase of Turing pattern formation without tissue growth, then start growth and observe the effects. Nevertheless, this first phase of tissue size stasis is not mandatory at all. Under particular conditions (notably if growth is not too fast compared to Turing dynamics), we could implement growth directly at the beginning, and PGCTPs could also arise. Also, concerning the growth rate dynamics, many different conditions (e.g. linear or exponential growth) can lead to the same final pattern. Then, I did not want to discuss all these possible conditions too much, but just to show that growth is sufficient to generate the patterns. 

To explain this choice, and to clarify things, I added explanations in the part 2, together with adding details in Figure 2 (see below in Minor revisions).

-> lines 270 to 273 “In all of our simulations, in order to simplify, we chose to let the Turing-like system to start in a non-growing domain, but it is not a mandatory condition. Then, at one point, growth starts, and we chose to make growth rate constant so the tissue surface grows exponentially, which is classical during development.”

In the 'Materials and Methods' section, particularly in the PGTCPs modeling session, it is imperative for the authors to provide essential information about the models. This should include the modeling assumptions, master equations for each circuit species, simulation methods, and expected outputs. Such details are crucial for comprehensively understanding and replicating the study's findings.

Thanks for this useful remark. Together with remarks of Reviewer 3, I decided to build a new model incorporating the effects of growth in diluting the concentrations of the agents. Therefore, I took this opportunity to describe more precisely the model, in the text in Materials and Methods. This information was previously available in the code, but now it is also available in the article text.

->lines 608 to 625 ”Master equations:

a(t+1)=(a(t))/(1+g*dt)+(16-(a(t))/(1+g*dt)*(b(t))/(1+g*dt) + (d(t))/((1+g*dt)*(1+g*dt) )*r*s*∇2a),; dt

b(t+1)=(b(t))/(1+g*dt)+((a(t))/(1+g*dt)*(b(t))/(1+g*dt) - (b(t))/(1+g*dt) – 12 + (d(t))/((1+g*dt)*(1+g*dt) )*s*∇2b)*dt

Where a and b are the morphogens concentrations, g is the growth rate (of the surface), r and s are ratio and scale parameters of De Gomensoro Malheiros et al equations (20), ∇2a and ∇2b the Laplacian operators. The discrete Laplacian operator used in the simulations is the standard nine-point stencil, covering a Moore neighborhood of unitary size: [ [1, 4, 1], [4, −20, 4], [1, 4, 1] ]/6.

The code of master equations:

 float newvaluea = oldvaluea/(1+g*dt) + (16 - oldvaluea*oldvalueb/((1+g*dt)*(1+g*dt)) + d*r*s*laplaceA)*dt;

 float newvalueb = oldvalueb/(1+g*dt) + (oldvaluea*oldvalueb/((1+g*dt)*(1+g*dt)) - oldvalueb/(1+g*dt) - 12 + d*s*laplaceB)*dt;

where g is the growth rate. Growth was indirectly implemented by its effect on diffusion (via s parameter) and its effect on morphogens concentration (via dilution, on parameters a and b).

g is equal to zero until a certain point (t = 10000 in all of our simulations), meaning that a classic Turing system starts to organize before growth starts.”

Besides, we add a table with all variables for each simulation presented in Figure. 2.

-> lines 633 to 634 “Variable parameters for simulations presented in this study:

Simulation Leopard Emu Pomacanthus Ictidomys Tetraodon

Initial conditions Random noise One middle stripe One middle stripe Periodic stripes Random noise

s 6 10 10 6 10

r 30 6 6 30 8

g 0.05 0.05 0.05 0.01 0.075

dt 0.002 0.002 0.002 0.002 0.002

Timing of

growth start 10000 10000 20000 1000 10000

“

Minor revisions:

In the figure legends, the authors should provide more comprehensive details about the key steps of each simulation experiment to facilitate a better understanding of the simulation results among readers.

Done. In every simulation presented in the Figure 2, I had details in every screenshot and in the Figure captions, in a complementary manner to the Material and Methods text.

Figure 1A:

Do these four categories encompass the entire spectrum of PGTCPs?

Thanks for this comment. They encompass all the spectrum of new geometries we described for PGTCPs.

For more clarity, I precise that these categories are exhaustive (in my current knowledge) in the Figure captions and in the plain text

-> lines 126 to 127: “The PGTCPs were classified in an exhaustive list of four categories,”

Figure 1C & 1D:

What criteria are used to differentiate classic Turing patterns from PGTCPs? Is this classification based on visual assessment of stable patterns in standard adult species? For instance, are spots, lines, and mazes classified as classic Turing patterns, while rosettes, line-and-dot alternations, mixed colors, and intermediate bands are categorized as PGTCPs?

Thanks for these questions. PGTCPs are all putative Turing-like patterns (i.e. periodic and not relying on pre-existing periodicity) which can not be reproduced by classic Turing simulations, whatever the system parameters are. So exactly: spots, lines and mazes are then classic Turing patterns, while rosettes, line-and-dot alternations, mixed colors, and intermediate bands are categorized as PGTCPs. 

Unfortunately, we cannot use adult stable patterns to define classic Turing patterns, because in most species the Turing system is stopped at one time during the development and the pattern is screenshot. For this reason, many adult “stable” patterns are PGTCPs “frozen in time”. 

Together with the first remark of Reviewer #1, this brought me to add a more complete and precise definition of PGTCPs in the beginning of the manuscript, in the introduction section:

-> lines 87 to 91 “Here, we consider a particular color pattern as a PGTCP when it possess 

---

## [Decision Letter · Decision Letter 1]

17 May 2024

PONE-D-23-43374R1And Growth on Form? How tissue expansion generates novel shapes, colours and enhance biological functions of Turing colour patterns of EukaryotesPLOS ONE

Dear Dr. Galipot,

Thank you for submitting your manuscript to PLOS ONE. After careful consideration, we feel that it has merit but does not fully meet PLOS ONE’s publication criteria as it currently stands. Therefore, we invite you to submit a revised version of the manuscript that addresses the points raised during the review process.

We look forward to receiving your revised manuscript.

Kind regards,

Hualin Fu

Academic Editor

PLOS ONE

Journal Requirements:

Additional Editor Comments:

Dr. Woolley raised an important point that the definition of PGTCP is not clear. In my opinion，the important message in this manuscript is that growth can modify or accelerate Turing patterning formation to create diverse morphologies in the Eukaryotes. In a simple word, growth is an important variable in the Turing patterning formation over time. I think the definition of PGTCP should be simplified and revised. It might be better to refer it simply as "growth accelerated Turing patterning".

Reviewers' comments:

Reviewer's Responses to Questions

**Comments to the Author**

1. If the authors have adequately addressed your comments raised in a previous round of review and you feel that this manuscript is now acceptable for publication, you may indicate that here to bypass the “Comments to the Author” section, enter your conflict of interest statement in the “Confidential to Editor” section, and submit your "Accept" recommendation.

Reviewer #1: (No Response)

2. Is the manuscript technically sound, and do the data support the conclusions?

Reviewer #1: Partly

3. Has the statistical analysis been performed appropriately and rigorously? 

Reviewer #1: N/A

4. Have the authors made all data underlying the findings in their manuscript fully available?

Reviewer #1: Yes

5. Is the manuscript presented in an intelligible fashion and written in standard English?

Reviewer #1: Yes

6. Review Comments to the Author

Reviewer #1: The author has addressed my comments, but I cannot say I'm satisfied by their respeonse.

My first criticism was a lack of definition. They have no added one, but if anything it strengthens my criticism. The definition is

"Here, we consider a particular color pattern as a PGTCP when it

possess all of these characteristics: i) periodic, at least at the local scale, ii) non-based

on a pre-existing periodicity (e.g. body segments or petals), iii) with at least three motifs

(e.g. three stripes or rosettes) and iv) with at least one motif not belonging to the

classical range of Turing geometries (dots, stripes, mazes), i.e. line-and-band

alternations, rosettes for example."

Firstly, this moves the problem onto defining what a motif is. Moreover, the first two are base properties of Turing patterns, (iii) is completely arbitrary (why three motifs?) and (iv) simply says it doesn't look like a Turing pattern. Nowhere in the definition is growth required, thus, many of the rebuttal points which are of the form: they don't include growth, so they're not PGTCPs, are invalid. In particular, most of the papers I've suggested in my previous review fit this description. It simply says that they're patterns that don't fit the normal Turing stereotypes, of which there is a lot of literature, which is currently being ignored.

Beyond that the conclusion is no stronger. The conclusion is still: Turing patterns can account for some patterns but not all. To achieve more complexity we need to add more complex mechanisms. This is known and the paper doesn't really add any conclusion beyond growth can change the produced Turing patterns.

7. PLOS authors have the option to publish the peer review history of their article (what does this mean?). If published, this will include your full peer review and any attached files.

Reviewer #1: No

---

## [Author Response · Author response to Decision Letter 1]

31 May 2024

Reviewer #1: The author has addressed my comments, but I cannot say I'm satisfied by their respeonse.

My first criticism was a lack of definition. They have no added one, but if anything it strengthens my criticism. The definition is

"Here, we consider a particular color pattern as a PGTCP when it

possess all of these characteristics: i) periodic, at least at the local scale, ii) non-based

on a pre-existing periodicity (e.g. body segments or petals), iii) with at least three motifs

(e.g. three stripes or rosettes) and iv) with at least one motif not belonging to the

classical range of Turing geometries (dots, stripes, mazes), i.e. line-and-band

alternations, rosettes for example."

Firstly, this moves the problem onto defining what a motif is. Moreover, the first two are base properties of Turing patterns, 

Thanks for your comment. The first two are indeed properties of Turing patterns, and it is not a coincidence, as this paper hypothesis is that PGTCPs are Turing-like patterns modified by growth (the “T” of PGTCP).

(iii) is completely arbitrary (why three motifs?)

I agree, this is partly arbitrary, but we chose to impose a minimum number of motifs for the following reason: when the number of motifs is low, there is a high probability that other types of mechanisms than Turing-like could produce them. For example, there is “no need” for a Turing system to produce a single stripe, we can find many other mechanisms in the litterature. We then decided to choose 3 motifs as a compromise, but we do not exclude the possibility that PGTCPs with only one or two motifs exist in nature, and alternatively that some of the three- or four-motifs patterns we have found might not be produced by a Turing-like system (This is why we chose the first “P” in PGTCPs, for Putative). For the same reason, we chose to exclude repeated patterns based on a pre-existed repeated geometry. For example, if every petal of a flower harbors a single dot, then of course the flower presents kind of a repeated color pattern, but in terms of probability that a Turing system produced it, it is the same “one-motif” case evoked earlier.

All the criteria were detailed in the Supp. Fig. 3 but we added some precision in the Figure legend and in the Material and Methods section.

lines 569 to 577: “We chose to impose a minimum number of motifs to consider it as a PGTCP for the following reason: when the number of motifs is low, there is a high probability that other types of mechanisms than Turing-like could produce them. We decided to choose 3 motifs as a compromise, but we do not exclude the possibility that PGTCPs with only one or two motifs exist in nature, and alternatively that some of the three- or four-motifs patterns we found might not be produced by a Turing-like system. For the same reason, we chose to exclude repeated patterns based on a pre-existed repeated geometry. For example, if every petal of a flower harbors a single dot and therefore an overall periodic color pattern, a Turing mechanism might not be needed to produce it.”

lines 1007 to 1009, in the Supp. Fig. 3 caption: “A minimal number of three motifs is required to consider it as a good candidate for a PGTCP, as one or two motifs patterns might be produced by other mechanisms than the Turing system.”

 and (iv) simply says it doesn't look like a Turing pattern. Nowhere in the definition is growth required, thus, many of the rebuttal points which are of the form: they don't include growth, so they're not PGTCPs, are invalid.

Thanks for this comment. Growth is not required in the definition because this analysis is based on pictures observation, so no data on growth rates are available (This is why we chose the first “P” in PGTCPs, for Putative). Of course, if we have access to the information that the tissue experiences no growth during pattern formation, we could exclude that this is a PGTCP. To my knowledge, this information is not yet available for any species bearing PGTCP, but to be honest, this is the subject of the next project I am working on.

 In particular, most of the papers I've suggested in my previous review fit this description. It simply says that they're patterns that don't fit the normal Turing stereotypes, of which there is a lot of literature, which is currently being ignored.

Beyond that the conclusion is no stronger. The conclusion is still: Turing patterns can account for some patterns but not all. To achieve more complexity we need to add more complex mechanisms. This is known and the paper doesn't really add any conclusion beyond growth can change the produced Turing patterns.

Thanks for this comment. To begin with, I agree that there is a lot of literature on non-classic Turing patterns, and much of it showed that growth is able to disrupt Turing systems and to generate non-canonical motifs. This is not the main subject of the article so it is normal that it doesn’t change the conclusions of the literature much. Nevertheless, in the modeling part summarized in Figure 2, we showed an alternative way on how growth can induce non-canonical motifs, which wasn’t -as far as I know- already described in the literature. Moreover, we explained how this or that effect of growth is adapted to this or that biological system (e.g. memory mechanism is adapted to the leopard, Fig. 2B, and the new mechanism is more adapted to the Pomacanthus Fig. 2D). In summary, growth could have effects but through several mechanisms. 

If our article was restricted to this modeling part, I would agree that “the paper doesn't really add any conclusion beyond growth can change the produced Turing patterns”.

However, the article presents many other aspects of Turing patterns and growth which go beyond this conclusion. Please let me rephrase the main conclusions of each part in a very summarized way:

Part 1/Figure 1: PGTCPs (i.e. patterns that might be produced by the interaction between growth and Turing systems) are present all over the Angiosperm and Animalia phylogenetic tree. From 4 or 5 species already described in animals, and no species mentioned on flowering plants, this article brings thousands of new species, representative of all the diversity. This large diversity could suggest specific biological functions for PGTCPs. 

Part 2/Figure 2: Modeling shows that growth could affect Turing patterns in many ways (including a novel one), and can account for all non-canonical Turing patterns in nature.

Part 3/Figure 3: PGTCPs occurrences analysis and morphometrics show that PGTCPs coincide with most-growing regions of the bodies, strongly suggesting that growth is responsible for the majority of them.

Part 4/Figure 4: Additionally to the biological functions of classic Turing color patterns, PGTCPs could allow individuals to be less visible (notably through mixing colors) and more visible (notably through the production of fractal motifs and with motifs not in the abiotic environment).

—-----------------------------

End of rebuttal letter

Pierre Galipot 

May, 31st 2024

---

## [Editor Report · Decision Letter 2]

7 Jun 2024

And Growth on Form? How tissue expansion generates novel shapes, colours and enhance biological functions of Turing colour patterns of Eukaryotes

PONE-D-23-43374R2

Dear Dr. Galipot,

We’re pleased to inform you that your manuscript has been judged scientifically suitable for publication and will be formally accepted for publication once it meets all outstanding technical requirements.

Kind regards,

Hualin Fu

Academic Editor

PLOS ONE

Additional Editor Comments (optional):

The revised manuscript is acceptable for publication.
---

## [Editor Report · Acceptance letter]

6 Sep 2024

PONE-D-23-43374R2 

PLOS ONE

Dear Dr. Galipot, 

I'm pleased to inform you that your manuscript has been deemed suitable for publication in PLOS ONE. Congratulations! Your manuscript is now being handed over to our production team.

Kind regards, 

on behalf of

Dr. Hualin Fu 

Academic Editor

PLOS ONE